# Mirabegron displays anticancer effects by globally browning adipose tissues

Xiaoting Sun[1,2,10], Wenhai Sui [3,10], Zepeng Mu[4], Sisi Xie[5], Jinxiu Deng[5], Sen Li[5], Takahiro Seki [2], Jieyu Wu [2], Xu Jing [2,6], Xingkang He[7], Yangang Wang[4], Xiaokun Li [1], Yunlong Yang [5] ✉, Ping Huang [8,9] ✉, Minghua Ge [6] ✉ & Yihai Cao [2] ✉

Metabolic reprogramming in malignant cells is a hallmark of cancer that relies on augmented glycolytic metabolism to support their growth, invasion, and metastasis. However, the impact of global adipose metabolism on tumor growth and the drug development by targeting adipose metabolism remain largely unexplored. Here we show that a therapeutic paradigm of drugs is effective for treating various cancer types by browning adipose tissues. Mirabegron, a clinically available drug for overactive bladders, displays potent anticancer effects in various animal cancer models, including untreatable cancers such as pancreatic ductal adenocarcinoma and hepatocellular carcinoma, via the browning of adipose tissues. Genetic deletion of the uncoupling protein 1, a key thermogenic protein in adipose tissues, ablates the anticancer effect. Similarly, the removal of brown adipose tissue, which is responsible for non-shivering thermogenesis, attenuates the anticancer activity of mirabegron. These findings demonstrate that mirabegron represents a paradigm of anticancer drugs with a distinct mechanism for the effective treatment of multiple cancers.

Cancer as a metabolic disease shows reprogrammed metabolic pathways to produce sufficient nutrients, energy molecules, metabolites, and growth signals to support ceaselessly growth and invasion[1,2]. Cancer cells preferably use glycolysis rather than oxidative phosphorylation for energy production by high glucose uptake and lactate production. Aerobic glycolysis also known as the Warburg effect signifies as proliferative metabolism is utilized by almost all cancer cells[3].

Consequently, targeting aerobic glycolysis by developing druggable inhibitors provides an attractive approach for cancer therapy. A number of preclinical and early clinical studies have demonstrated the effectiveness of antiglycolytic agents[4].

Adipose tissues constitute one of the largest tissue compositions in the body and maintain energy balance by storage of excessive energy as lipid droplets or expenditure of energy by

[1]Oujiang Laboratory (Zhejiang Lab for Regenerative Medicine, Vison and Brain Health), School of Pharmaceutical Science, Wenzhou Medical University, Wenzhou, China. [2]Department of Microbiology, Tumor and Cell Biology, Karolinska Institutet, 171 65 Solna, Sweden. [3]National Key Laboratory for Innovation and Transformation of Luobing Theory National Key Laboratory for Innovation and Transformation of Luobing Theory; The Key Laboratory of Cardiovascular Remodeling and Function Research, Chinese Ministry of Education, Chinese National Health Commission and Chinese Academy of Medical Sciences, The State and Shandong Province Joint Key Laboratory of Translational Cardiovascular Medicine, Department of Cardiology, Qilu Hospital of Shandong University, 250012 Jinan, China. [4]Department of Endocrinology, Affiliated Hospital of Medical College Qingdao University, Qingdao, China. [5]Department of Cellular and Genetic Medicine, School of Basic Medical Sciences, Fudan University, 200032 Shanghai, China. [6]Department of Head and Neck Surgery, Center of Otolaryngology-Head and Neck Surgery, Zhejiang Provincial People's Hospital, People's Hospital of Hangzhou Medical College, Hangzhou, China. [7]Department of Gastroenterology, Sir Run Run Shaw Hospital, Zhejiang University Medical School, Hangzhou 310016, China. [8]Department of Pharmacy, Zhejiang Provincial People's Hospital, People's Hospital of Hangzhou Medical College, Hangzhou, China. [9]Key Laboratory of Endocrine Gland Diseases of Zhejiang Province, Hangzhou, China. [10]These authors contributed equally: Xiaoting Sun, Wenhai Sui. ✉e-mail: yunlongyang@fudan.edu.cn; huangping@hmc.edu.cn; geminqh@163.com; yihai.cao@ki.se

lipolysis. While adult white adipose tissues (WATs) store excessive energy and undergo relentlessly expansion and shrinkage depending on metabolic status, brown adipose tissue (BAT) mainly entails energy expenditure by thermogenic metabolism[5,6]. Under cold exposure, certain drug treatments, special diets, and pathological conditions, WAT experiences both phenotypic and metabolic reprogramming to become a brown-like tissue i.e., browning WAT. Adipocytes in browning WAT contain decreased lipid droplets, increased mitochondrial contents, and smaller adipocyte sizes, which resemble metabolically active BAT adipocytes[7]. Accumulative research evidence demonstrates that activation of BAT and browning WAT is an effective approach for energy consumption and beneficial for improving metabolic disorders such as type 2 diabetes[8,9]. BAT and WAT browning involves sympathetic activation of the β3-adrenergic receptor (β3-AR). An adult human possesses significant amounts of BAT in supraclavicular, paravertebral, and perirenal areas, which can be activated by cold exposure or a β3-AR agonist[10–12].

Activated browning adipocytes, especially those in BAT, show high glucose intake, expression of numerous browning markers, and mitochondrial contents. In particular, uncoupling protein 1 (UCP1), a mitochondrial carrier protein located in the inner membrane of mitochondria and a key protein for non-shivering thermogenesis (NST), becomes highly upregulated in adipocytes of activated BAT and browning WAT[13]. Despite marked increases in glucose uptake in browning adipose tissues, the involvement of glucose in thermogenesis requires further investigation[14,15].

Aside from driving thermogenesis, our group recently reported that cold exposure is a potent methodology for inhibiting solid tumor growth via BAT activation and energy expenditure[15], which provides a distinct concept for cancer treatment. Although various studies show that cold-induced thermogenesis has pronounced effects on glucose homeostasis, the majority of the animal studies were conducted under 4 °C cold exposure, a condition that cannot be sustained for long periods of time in clinical settings. In humans, it seems like many factors influence cold-induced NST, such as body composition[16] and ageing[17], making it difficult to determine the suitable temperature for cold therapy. This prompted us to investigate the anticancer efficacy of other clinical relevant BAT activation methods besides cold, such as drugs. Currently, there are no reports of anticancer activity in BAT activation drugs. Moreover, whether these drugs have the same mechanism and effect as cold exposure is not explored.

Mirabegron is the first clinically available beta-3 agonist with approval for the treatment of overactive bladder in human patients[18]. Chronic treatment of human subjects with mirabegron markedly increases brown adipose tissue by enhanced uptake of [18]F-2-fluoro-d-2-deoxy-d-glucose ([18]F-FDG) scanned by positron emission tomography-computed tomography (PET-CT). Resting energy expenditure (REE) is higher in mirabegron-treated individuals relative to controls without altering bodyweight and composition[8]. Importantly, chronic mirabegron treatment increases insulin sensitivity and glucose utilization[8]. These human studies support the concept of activation of BAT by a β3-AR for the treatment of metabolic disease.

In this study, we propose a distinct concept of anticancer therapy by a drug that induces browning adipose tissue. In several mouse cancer models, we provide compelling evidence that mirabegron displays broad and potent anticancer effects on various cancers. Mirabegron´s anticancer effect is completely dependent on the browning of adipose tissues, which includes both BAT and WATs. Genetic deletion of *Ucp1* in mice or surgical removal of adipose depots abrogated tumor suppression. On the basis of these findings, we define a paradigm of anticancer drugs by instigating browning of adipose tissues.

## Results

### Mirabegron inhibits tumor growth and prolongs survival in various tumor models

To study the effect of mirabegron-induced metabolic changes in adipose tissues on tumor growth and progression, mirabegron at a dose of 8 mg/kg, which was known to instigate adipose tissue browning in our previous studies[19], was administrated once daily in various tumor-bearing animal models. Pancreatic ductal adenocarcinoma (PDAC) is the most lethal malignancy and lacks effective treatment[20]. To test mirabegron's potential anticancer effect on PDAC, murine Panc02 pancreatic cancer cells were subcutaneously implanted in wild-type C57BL/6 mice. Interestingly, mirabegron treatment did not alter tumor cell growth in vitro (Fig. S1a), but resulted in a more than 50% tumor growth rates reduction in vivo (Fig. 1a, Fig. S1b and c). We further investigated the impact of different dosages of mirabegron on the reduction of tumor growth rate, and found that the anticancer effect was dose-dependent. The dose as low as 3.2 mg/kg was effective to reduce tumor growth rate (Fig. S1d). In line with suppressed tumor growth, the survival of PDAC tumor-bearing mice was markedly prolonged upon mirabegron treatment (Fig. 1b). In tumor tissue, Ki67-measured proliferative rate of tumor cells was substantially inhibited, while cleaved caspase-3-measured apoptosis was unaltered. Consistent with the suppressed tumor proliferation, mirabegron-treated PDAC tumors exhibited alleviated hypoxia (Fig. 1c). To generalize these findings, we investigated mirabegron's effects in another hard-to-treat cancer type. Clinically, treatment of advanced hepatocellular carcinoma (HCC) remains challenging, with very few drugs approved[21]. We established an HCC orthotopic model with Hepa1-6 cells surgically implanted in the liver and treated with vehicle or mirabegron. Similarly, significant reductions were observed in tumor growth, tumor proliferation, and tumor hypoxia in HCC tumors, and mice exhibited a prolonged overall survival (Fig. 1d, e). To further corroborate mirabegron's anticancer effect, mouse colorectal cancer (CRC) subcutaneous model and genetic spontaneous *Apc*[Min/+] intestinal adenoma models were applied. Similarly, mirabegron strongly suppressed tumor growth and tumor formation in these models (Figs. 1f and S1e-f). These results show that mirabegron suppresses solid tumor growth in various cancer types.

### Mirabegron-induced adipose tissue browning and glucose uptake pattern shifts

To test whether mirabegron activates adipose tissue browning in tumor-bearing mice, we investigated the BAT, subcutaneous WAT (subWAT), and visceral WAT (visWAT) upon approximately 2-week mirabegron treatment in PDAC tumor-bearing mice. Expectedly, similar to tumor-free mice[19], mirabegron treatment increased smaller multilocular structures in BAT, suggesting BAT activation (Fig. 2a). Histological analysis of adipose depots supported a browning phenotype in mirabegron-treated WATs, in which adipocyte sizes were significantly smaller in mirabegron-treated adipose tissues compared with their controls (Fig. 2a). In particular, visWAT appeared as high-density multilocular structures, suggesting visWAT browning (Fig. 2a). Consistently, UCP1 and the cytochrome c oxidase subunit 4 (COX4)+ mitochondrion contents were increased in mirabegron-treated adipose depots (Fig. 2a). These results were further validated in HCC orthotopic model, CRC model, and genetic spontaneous *Apc*[Min/+] intestinal adenoma model (Fig. S2a-c), indicating that mirabegron activates adipose tissue browning in various adipose depots in tumor-bearing mice.

It is known that adipose tissue browning mediates NST via UCP1 in the mitochondrial inner membrane and improves metabolic dysfunctions via intense energy-expenditure. We hypothesized that the anticancer effect of mirabegron is energy-expenditure dependent. To link the adipose tissue browning and tumor inhibition, we

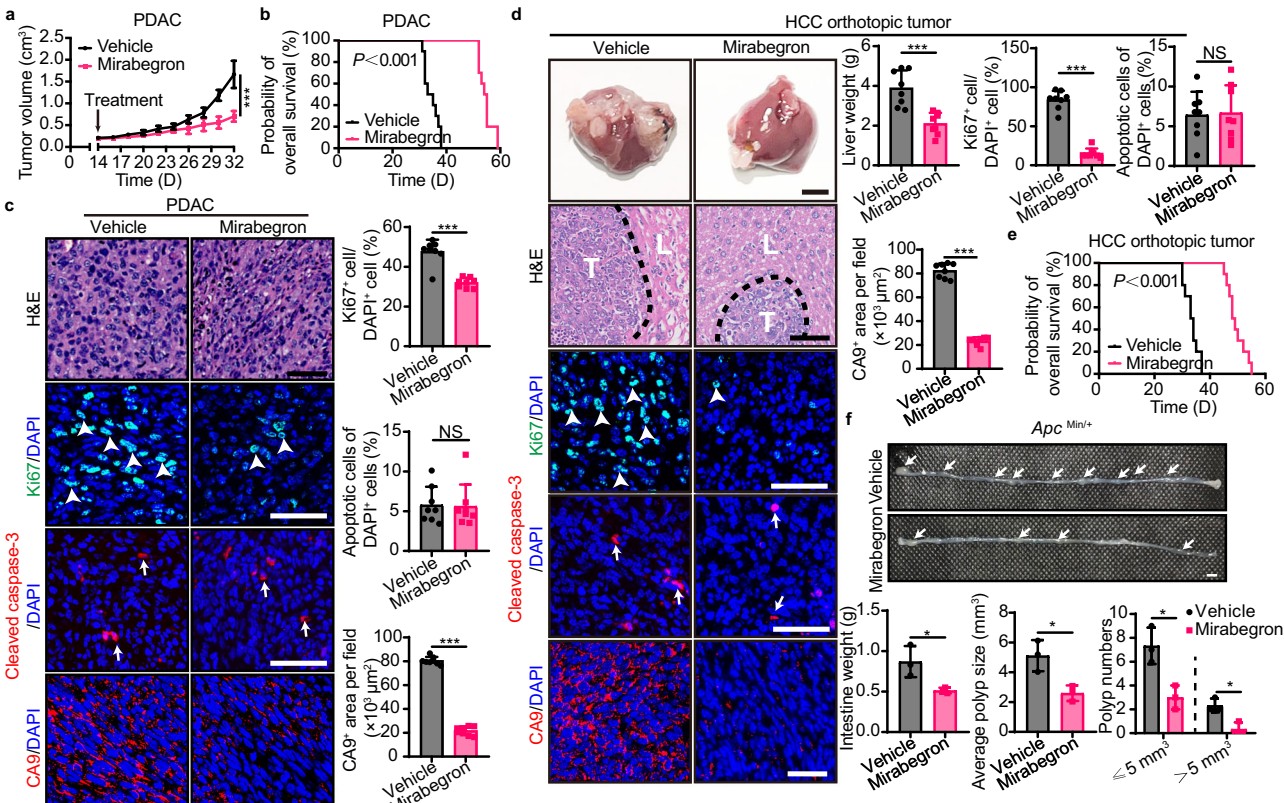

**Fig. 1 | Mirabegron inhibits tumor growth and prolongs survival in tumor-bearing mice. a** Tumor growth of vehicle- and mirabegron-treated PDAC tumor-bearing mice ($n = 8$ mice per group). **b** Overall survival of vehicle- and mirabegron-treated PDAC tumor-bearing mice ($n = 10$ mice per group). **c** H&E histological staining and immunofluorescence staining of Ki67$^+$ proliferating cells (green), cleaved-caspase 3$^+$ apoptotic cells (red), and CA9$^+$ hypoxic area (red) of PDAC tumors. Tissues were counterstained with 4′,6-diamidino-2-phenylindole (DAPI, blue). Arrows and arrowheads point to their respective positive signals. Scale bar, 50 μm. Quantifications of Ki67$^+$ signal, cleaved-caspase 3$^+$ signal, and CA9$^+$ signal ($n = 8$ random fields per group). **d** Liver photographs and liver weights of vehicle- and mirabegron-treated orthotopic HCC tumor-bearing mice ($n = 8$ mice per group). H&E histological staining and immunofluorescence staining of Ki67$^+$ proliferating cells (green), cleaved-caspase 3$^+$ apoptotic cells (red), and CA9$^+$ hypoxic area (red) of HCC tumors. Tissues were counterstained with DAPI (blue). Arrows

and arrowheads point to their respective positive signals. T, tumor. L, liver. Scale bar in upper panel, 5 mm. Scale bar in lower four panels, 50 μm. Quantifications of Ki67$^+$ signal, cleaved-caspase 3$^+$ signal, and CA9$^+$ signal ($n = 8$ random fields per group). **e** Overall survival of vehicle- and mirabegron-treated orthotopic HCC tumor-bearing mice ($n = 10$ mice per group). **f** Photographs and weights of intestine in vehicle- and mirabegron-treated $Apc^{Min/+}$ mice ($n = 3$ mice per group). Quantifications of polyp distribution and average sizes in vehicle- and mirabegron-treated $Apc^{Min/+}$ mice ($n = 3$ mice per group). Scale bar, 5 mm. Statistical analysis was performed using two-sided unpaired t-test (**a**, **c**, **d**, **f**) and log-rank test (**b**, **e**). NS, not significant; *$P < 0.05$; **$P < 0.01$; ***$P < 0.001$. Data presented as mean ± SD. Each experiment was repeated at least three times and the representative experiment was shown (**a**–**e**). $Apc^{Min/+}$ mice experiment was performed once. Source data are provided as a Source Data file.

performed PET-CT imaging analysis by measuring the uptake of $^{18}$F-FDG in PDAC tumor-bearing mice. In PDAC tumor-bearing mice with approximately equal tumor size, mirabegron stimulation strongly induced $^{18}$F-FDG uptake in BAT. Notably, WAT also showed an increase in $^{18}$F-FDG uptake. Interestingly, $^{18}$F-FDG signals were dramatically inhibited in PDAC tumors (Fig. 2b), suggesting a glucose uptake paradigm shift in mirabegron-treated tumor-bearing mice. To investigate whether the browning process improved metabolic health, we tested blood insulin, blood glucose, insulin tolerance, and glucose tolerance in tumor-bearing mice with or without mirabegron treatment. Fast blood insulin and glucose levels were significantly decreased and insulin tolerance and glucose tolerance were markedly improved in mirabegron-treated groups (Fig. 2c–f). Furthermore, mice treated with mirabegron exhibited high metabolic rates measured by $O_2$ consumption and $CO_2$ production, suggesting adipose tissue browning-mediated global metabolism upon mirabegron treatment (Fig. 2g and h). Together, these findings indicate that mirabegron reduces glucose uptake in tumors by glucose uptake pattern shift towards adipose depots.

Mirabegron induces metabolic effects mainly through its β3 agonist ability[22]. To study the role of β3 signaling in the anticancer

effect of mirabegron, we head-to-head compared mirabegron, isoproterenol, a non-specific β-AR agonist, and CL-316, 243, a widely used β3-specific agonist, in CRC model in vitro and in vivo. Similar to mirabegron, neither isoproterenol nor CL-316, 243 treatment altered tumor cell growth in vitro (Fig. S3a), but they all inhibited tumor growth in vivo to a similar extent as mirabegron (Fig. S3b). β-AR antagonists, including propranolol, L-748,337, and SR59230A, largely alleviated the antitumor effect of mirabegron (Fig. S3c). In our study, to mimic clinical settings, mirabegron is administered at room temperature (about 22 °C), which has been reported to induce chronic cold stress. To investigate the potential effect of room temperature on adipose tissue browning and tumor inhibition, we administered mirabegron to tumor-bearing mice in climate chambers of 22 °C and thermoneutrality (30 °C). The results showed that there was no difference in tumor growth between room temperature and thermoneutrality (Fig. S3d). Furthermore, we examined the $Ucp1$ expression of BAT and subWAT under these two conditions. Again, no difference was observed between these two groups (Fig. S3e). These results support that the anticancer effect of mirabegron is related to the β3-induced adipose tissue browning.

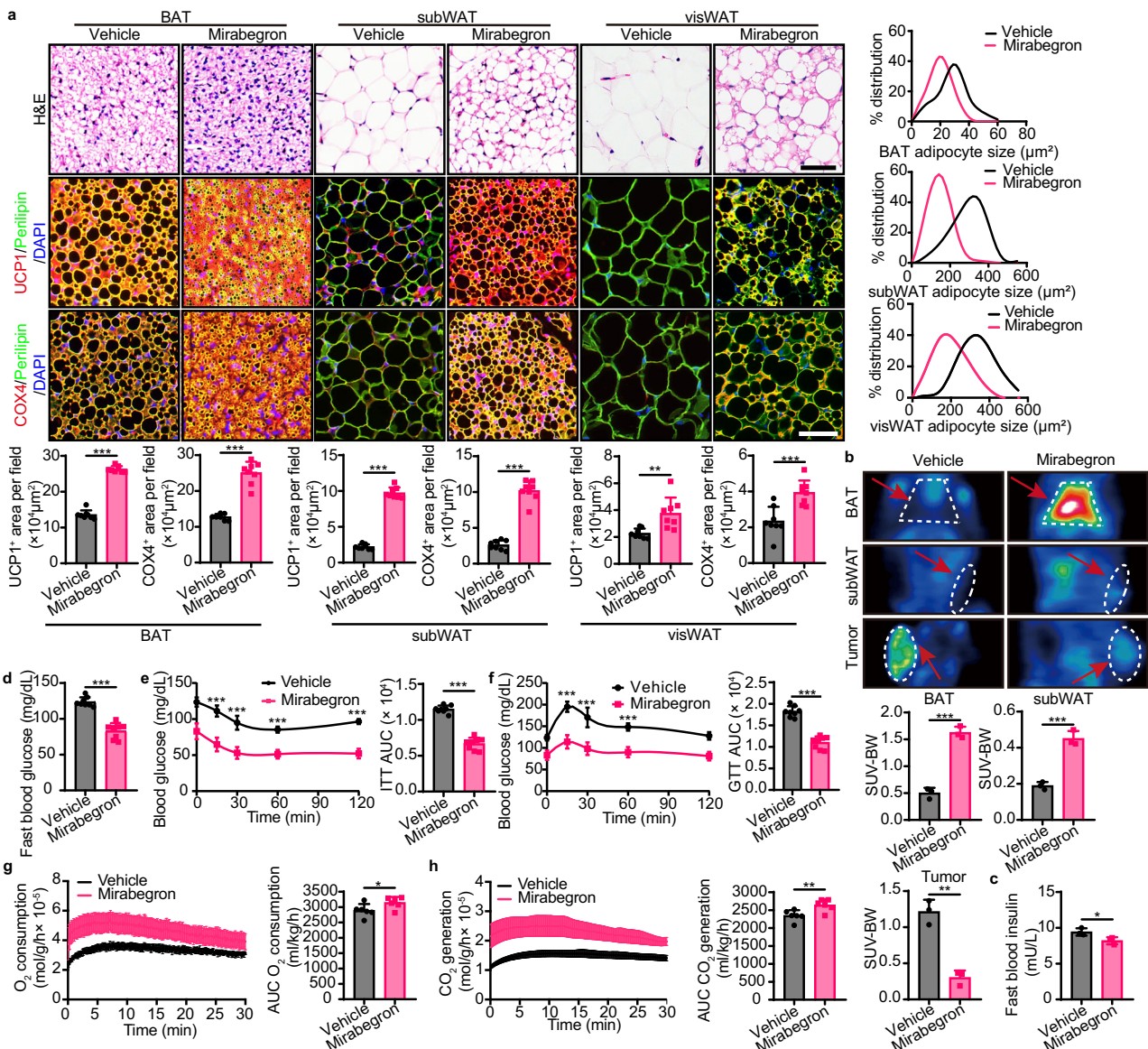

**Fig. 2 | Adipose activation and glucose uptake in adipose tissues and tumors.** **a** H&E histological staining and immunofluorescence staining of UCP1 (red), COX4 (red), and Perilipin (green) of adipose depots in vehicle- and mirabegron-treated PDAC tumor-bearing mice. Tissues were counterstained with DAPI (blue). Scale bar, 50 μm. Quantifications of adipocyte size distribution, UCP1+ signal, and COX4+ signal (n = 8 random fields per group). **b** Representative PET-CT images of BAT, subWAT, and tumors in vehicle- and mirabegron-treated PDAC tumor-bearing mice. Quantification of standardized uptake values (SUV) justified by body weight (SUV-BW) of BAT, subWAT, and tumors (n = 3 mice per group). **c, d** Fast blood insulin levels and glucose levels of vehicle- and mirabegron-treated PDAC tumor-

bearing mice (n = 3 mice per group/ 8 mice per group). **e, f** Insulin tolerance test (ITT) and glucose tolerance test (GTT) of vehicle- and mirabegron-treated PDAC tumor-bearing mice (n = 8 mice per group). **g, h** Whole-body metabolism measured by $O_2$ consumption and $CO_2$ generation in vehicle- and mirabegron-treated PDAC tumor-bearing mice (n = 6 mice per group). Statistical analysis was performed using two-sided unpaired t-test (a-h). NS, not significant; *$P < 0.05$; **$P < 0.01$; ***$P < 0.001$. Data presented as mean ± SD. Each experiment was repeated at least three times and the representative experiment was shown (**a**, **c**–**f**). PET-CT test (**b**) and whole-body metabolism (**g**, **h**) were performed once. Source data are provided as a Source Data file.

## Both BAT and WAT contribute to the mirabegron-instigated tumor suppression

To further investigate the role of BAT activation in tumor suppression, we next surgically removed BAT in PDAC tumor-bearing mice. In vehicle-treated groups, removal of BAT did not alter PDAC tumor growth (Fig. 3a). Compared with the sham operation group, removal of BAT significantly alleviated mirabegron-inhibited tumor growth (Fig. 3a). BAT removal improved fast blood glucose, insulin, c-peptide, tumor cell proliferation, and tumor hypoxia, favoring tumor growth (Fig. 3b–d). Of note, BAT removal did not completely abolish mirabegron's anticancer effect (Fig. 3a). Considering that both subWAT and visWAT occupy a large volume of WAT, their

browning may also affect glucose uptake and consequently alter global glucose metabolism. To test the role of white-to-beige transition in tumor suppression, we performed a WAT removal surgery in PDAC tumor-bearing mice. Interestingly, removal of these WAT depots partially alleviated mirabegron-inhibited tumor growth, and improved tumor cell proliferation and tumor hypoxia (Fig. S4a-c). Next, we removed BAT, subWAT, and visWAT in PDAC tumor-bearing mice. Again, removal of all three depots did not alter tumor growth (Fig. 3e). Interestingly, upon mirabegron treatment, removal of all three adipose depots abolished the blood glucose inhibition effect and nearly completely abolished tumor-suppression effect in mirabegron-treated mice (Fig. 3e–g). These

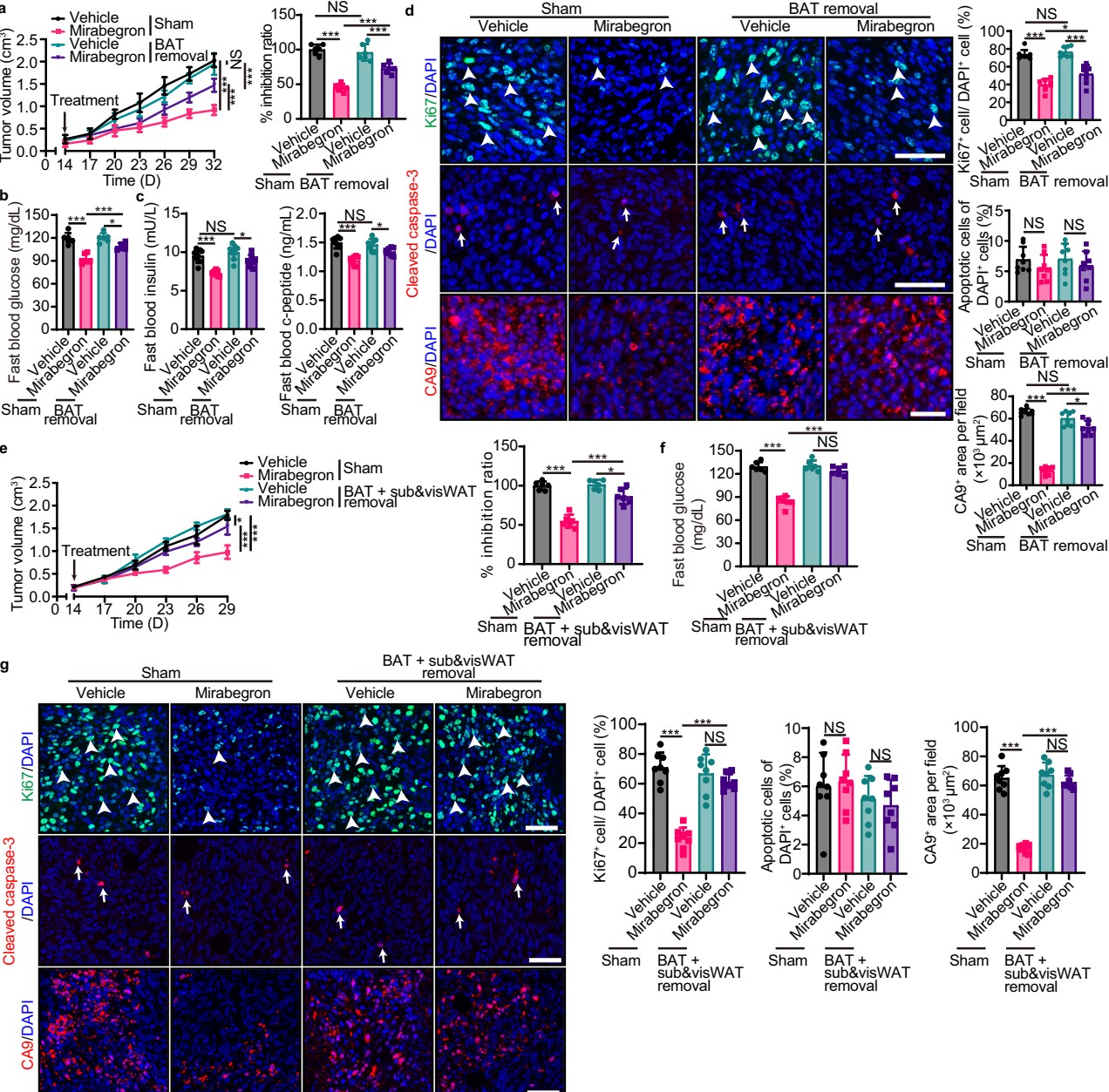

**Fig. 3 | Removal of adipose tissues ablates mirabegron-triggered tumor suppression. a** PDAC tumor growth and inhibition ratios of sham and BAT removal tumor-bearing mice with or without mirabegron treatment (*n* = 6 mice per group). **b** Fast blood glucose levels of sham and BAT removal tumor-bearing mice with or without mirabegron treatment (*n* = 6 mice per group). **c** Fast blood insulin and c-peptide levels of sham and BAT removal tumor-bearing mice with or without mirabegron treatment (*n* = 8 mice per group). **d** Immunofluorescence staining of Ki67+ proliferating cells (green), cleaved-caspase 3+ apoptotic cells (red), and CA9+ hypoxic area (red) of PDAC tumors. Tissues were counterstained with DAPI (blue). Arrows and arrowheads point to their respective positive signals. Scale bar, 50 μm. Quantifications of Ki67+ signal, cleaved-caspase 3+ signal, and CA9+ signal (*n* = 8 random fields per group). **e** PDAC tumor growth and inhibition ratios of sham and fat depots (BAT, subWAT, and visWAT) removal tumor-bearing mice with or without mirabegron treatment (*n* = 6 mice per group). **f** Fast blood glucose levels of sham and fat depots removal tumor-bearing mice with or without mirabegron treatment (*n* = 6 mice per group). **g** Immunofluorescence staining of Ki67+ proliferating cells (green), cleaved-caspase 3+ apoptotic cells (red), and CA9+ hypoxic area (red) of PDAC tumors. Tissues were counterstained with DAPI (blue). Arrows and arrowheads point to their respective positive signals. Scale bar, 50 μm. Quantifications of Ki67+ signal, cleaved-caspase 3+ signal, and CA9+ signal (*n* = 8 random fields per group). Statistical analysis was performed using one-way ANOVA test (**a**–**g**). NS, not significant; *$P < 0.05$; **$P < 0.01$; ***$P < 0.001$. Data presented as mean ± SD. Each experiment was repeated at least three times and the representative experiment was shown (**a**–**g**). Source data are provided as a Source Data file.

results support both BAT's and WAT's role in mirabegron-instigated tumor suppression.

## Mirabegron induces metabolic reprogramming in tumors

The tight link between adipose tissue browning and tumor growth leads us to explore the mirabegron-instigated metabolic changes in

tumors. To identify the potential metabolic process in mirabegron-treated tumors, tumor tissues were projected for unbiased metabolomic analysis, and both the differentially expressed metabolites analysis and enrichment analysis were conducted. Principal component analysis (PCA) confirmed the metabolic separation between vehicle- or mirabegron-treated PDAC tumors (Fig. S5a). The variable importance

in projection (VIP) score was applied to determine the separation potential of the metabolite. As a result, 96 metabolites were shown to be differentially expressed based on our predefined criteria (p < 0.05, VIP > 1, and |log2 (fold change)| > 1) (Fig. S5b). Annotated biological processes of these metabolites were summarized by KEGG enrichment analysis (Fig. 4a). Consistent with the reduced [18]F-FDG uptake in mirabegron-treated tumors, the central carbon metabolism pathway was enriched in the comparison (Fig. 4a). We next performed a heatmap of selected carbohydrate metabolism-related metabolites in the differentially expressed metabolites. Interestingly, the abundance of these metabolites was suppressed (Fig. 4b), supporting the jeopardized glucose metabolism in tumor tissues. Of note, L-glutamine

was dramatically reduced and L-glutamate was increased in the mirabegron-treated tumors (Fig. 4b). Most cancer cells depend on glutamine for accelerating the tricarboxylic acid cycle by converting to α-ketoglutarate via glutamate[23]. The upregulated L-glutamate/L-glutamine suggests that tumors may be more dependent on glutamine metabolism under mirabegron-induced glucose deprivation. To validate the inhibited glucose metabolism pathway, we detected the RNA expression levels of key glucose metabolism-related genes. In tumor tissue, glucose transporter (GLUT) 1, 4, 7 (*Glut1*, *Glut4*, and *Glut7*), phosphoglucomutase 1, 2 (*Pgm1*, *Pgm2*), hexokinase 1, 2, 3 (*Hk1*, *Hk2*, *Hk3*), pyruvate kinase M1/2 (*Pkm*), and phosphofructokinases (*Pfkp*, *Pfkl*) were significantly inhibited (Fig. 4c), supporting the reduction of

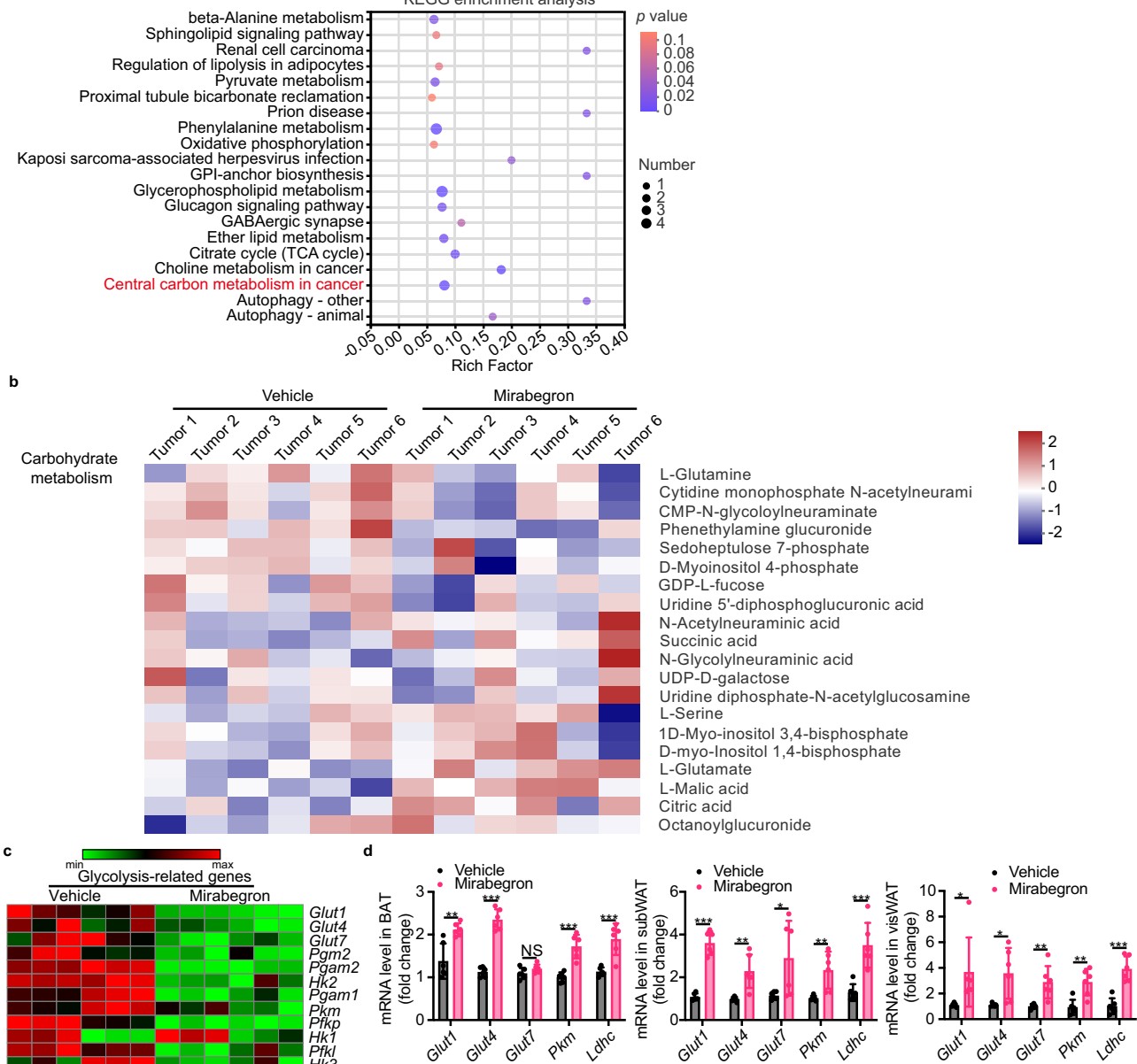

**Fig. 4 | Mirabegron inhibits glucose metabolism in tumor tissues. a** Metabolic pathway enrichment analysis of differentially expressed metabolites between vehicle- and mirabegron-treated PDAC tumors (*n* = 6 samples per group). Overview of enriched metabolic pathways in the mirabegron-treated tumors compared to the vehicle-treated tumors. Pathway impact was analyzed using MetaboAnalyst 3.0. **b** Heatmaps of the differential carbohydrate metabolism-related metabolites (*n* = 6 samples per group). **c.** Heatmaps of GLUTs and glycolysis-related genes in vehicle- and mirabegron-treated PDAC tumors (*n* = 6 samples per group). **d** qPCR

analysis of GLUTs and glycolysis-related genes in BAT, subWAT, and visWAT in vehicle- and mirabegron-treated PDAC tumor-bearing mice (*n* = 6 samples per group). Statistical analysis was performed using two-sided unpaired t-test (**d**). *P* value < 0.05 was considered significant enrichment (**a**). NS, not significant; **P* < 0.05; ***P* < 0.01; ****P* < 0.001. Data presented as mean ± SD. Each experiment was repeated at least three times and the representative experiment was shown (**d**). Metabolomic-related analysis (**a**–**c**) were performed once. Source data are provided as a Source Data file.

glucose metabolism in tumor tissues. In contrast, these critical genes were significantly upregulated in BAT, subWAT, and visWAT (Fig. 4d), suggesting the shift of glucose uptake and glucose metabolism between tumor and adipose depots.

In addition to glucose, we detected metabolic pathways for other energy supplies. Reduction patterns were evident in metabolites related to lipid metabolism (Fig. S5c). Fatty acid oxidation-related genes and free fatty acid transporter-related genes were downregulated in tumor tissues after mirabegron treatment (Fig. S5d and e). These results suggest that mirabegron-induced NST results in the downregulation of multiple metabolic pathways in tumors.

### High-glucose restores tumor growth

To investigate whether the glucose metabolic shift between tumor and adipose depots is the underlying mechanism for mirabegron-induced tumor suppression, we tested high-glucose feeding, a 15% glucose in the drinking water ad libitum, in vehicle- or mirabegron-treated tumor-bearing mice. Compared to normal chow (Fig. 1a), high-glucose feeding significantly increased non-fast blood glucose levels, without changing the fast blood glucose levels (Fig. 5a). Expectedly, feeding 15% glucose completely abolished the mirabegron's anticancer effect compared to control groups (Fig. 5b), with tumor proliferation and tumor hypoxia restored (Fig. 5c). Interestingly, 15% glucose only partially restored the mirabegron-reduced WAT weight (Fig. 5d), suggesting the supplement of carbohydrates does not fully compensate for the browning-instigated lipolysis in tumor-bearing mice. These results suggest that high-glucose feeding restores tumor growth upon mirabegron treatment.

### Mirabegron-instigated tumor suppression is UCP1-dependent

The critical thermogenerator protein in NST is UCP1, which is expressed in both brown adipocytes and beige adipocytes. To test thermogenesis as the energy expenditure mechanism for mirabegron-suppressed tumor, we applied $Ucp1^{-/-}$ mice in C57BL/6 background for tumor implantation. Deletion of the $Ucp1$ gene abolished mirabegron's anticancer effect compared to WT mice (Fig. 6a), with the unaltered

fast blood glucose, insulin, c-peptide, insulin tolerance, and glucose tolerance (Fig. 6b–e). Similarly, $Ucp1^{-/-}$ restored tumor cell proliferation and hypoxia (Fig. 6f). Furthermore, tumor glucose metabolism-related gene expressions were largely restored in $Ucp1^{-/-}$ mice (Fig. 6g and h). These results support that UCP1-mediated glucose metabolism shifts are essential for mirabegron-instigated tumor suppression.

## Discussion

Based on their targets, anticancer drugs can be divided into three categories: 1) Malignant cell-targeted therapeutics; 2) Tumor stroma-targeted agents; and 3) Dual cancer cell- and stroma-targeted therapeutics such as chemotherapy. Targeted therapeutics usually are specifically effective for a particular cancer type that expresses a specific marker or overexpresses a drug-targeted molecule. Because malignant cells in a tumor consist of genetically diverse heterogeneous populations and contain unstable genomes with various mutations, targeted anticancer agents likely experience drug resistance after prolonged therapy[24]. Moreover, treatment with targeted therapeutics would impose a selection of cancer cell populations that can survive under drug pressures[25]. Consequently, resistance tumors become completely insensitive to the original targeted therapy. Unlike targeted anticancer agents, those drugs that target the stromal components in the tumor microenvironment (TME) such as antiangiogenic drugs (AADs) and immune checkpoint inhibitors (ICIs) employ generalized mechanisms for cancer therapy that can be used for the treatment of various types of cancer. For example, AADs and ICIs have received FDA approval for treatment of a myriad of cancer types in human patients[26–28]. Despite targeting genetically stable stromal components by AADs and ICIs, these drugs often encounter resistance through complex mechanisms[29,30]. Development of drug resistance and TME heterogeneity in various cancers have mitigated the clinical benefits of these drugs. Together, currently available anticancer drugs in the clinic are aimed to eliminate cancer cells or alter TME.

Aerobic glycolysis-dependent growth of malignant cells has provided opportunities for the development of drugs to metabolically retard tumor growth. Most therapeutic approaches that target

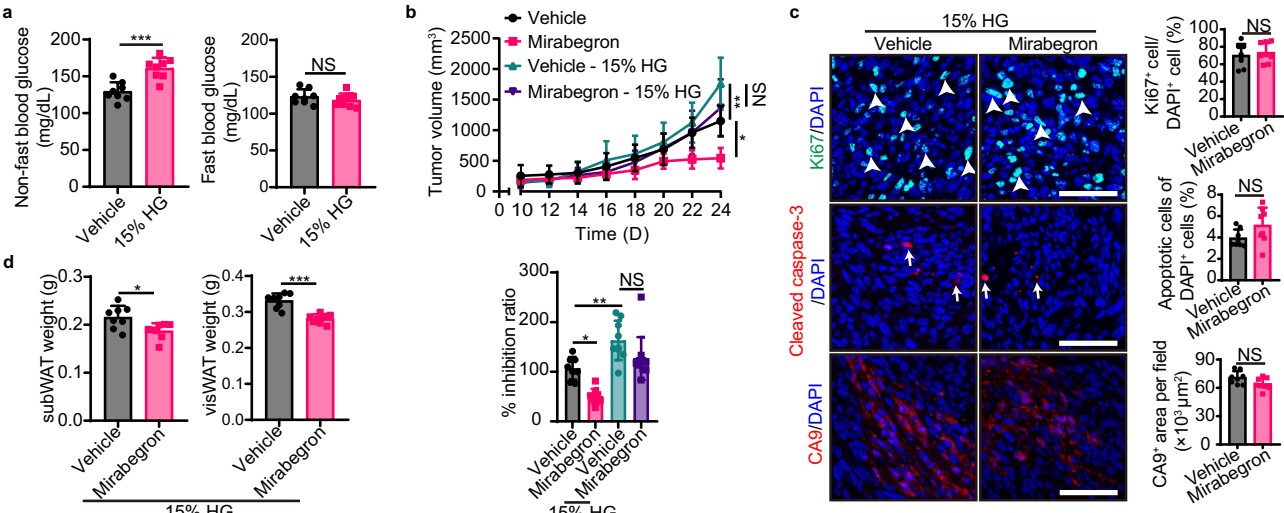

**Fig. 5 | High glucose feeding abrogates mirabegron-induced tumor suppression. a** Fast blood insulin and non-fast blood insulin levels of vehicle- and 15% glucose-treated PDAC tumor-bearing mice (*n* = 8 mice per group). **b** PDAC tumor growth and inhibition ratios of vehicle and mirabegron-treated tumor-bearing mice receiving vehicle or 15% glucose feeding (*n* = 8-11 mice per group). **c** Immunofluorescence staining of Ki67+ proliferating cells (green), cleaved-caspase 3+ apoptotic cells (red), and CA9+ hypoxic area (red) of PDAC tumors. Tissues were counterstained with DAPI (blue). Arrows and arrowheads point to their respective

positive signals. Scale bar, 50 μm. Quantifications of Ki67+ signal, cleaved-caspase 3+ signal, and CA9+ signal (*n* = 8 random fields per group). **d** White adipose tissue weight of vehicle- and mirabegron-treated tumor-bearing mice receiving 15% glucose feeding (*n* = 8 mice per group). Statistical analysis was performed using two-sided unpaired t-test (**a**, **c**, **d**), one-way ANOVA test (**b**). NS, not significant; *$P$ < 0.05; **$P$ < 0.01; ***$P$ < 0.001. Data presented as mean ± SD. Each experiment was repeated at least three times and the representative experiment was shown (**a**–**d**). Source data are provided as a Source Data file.

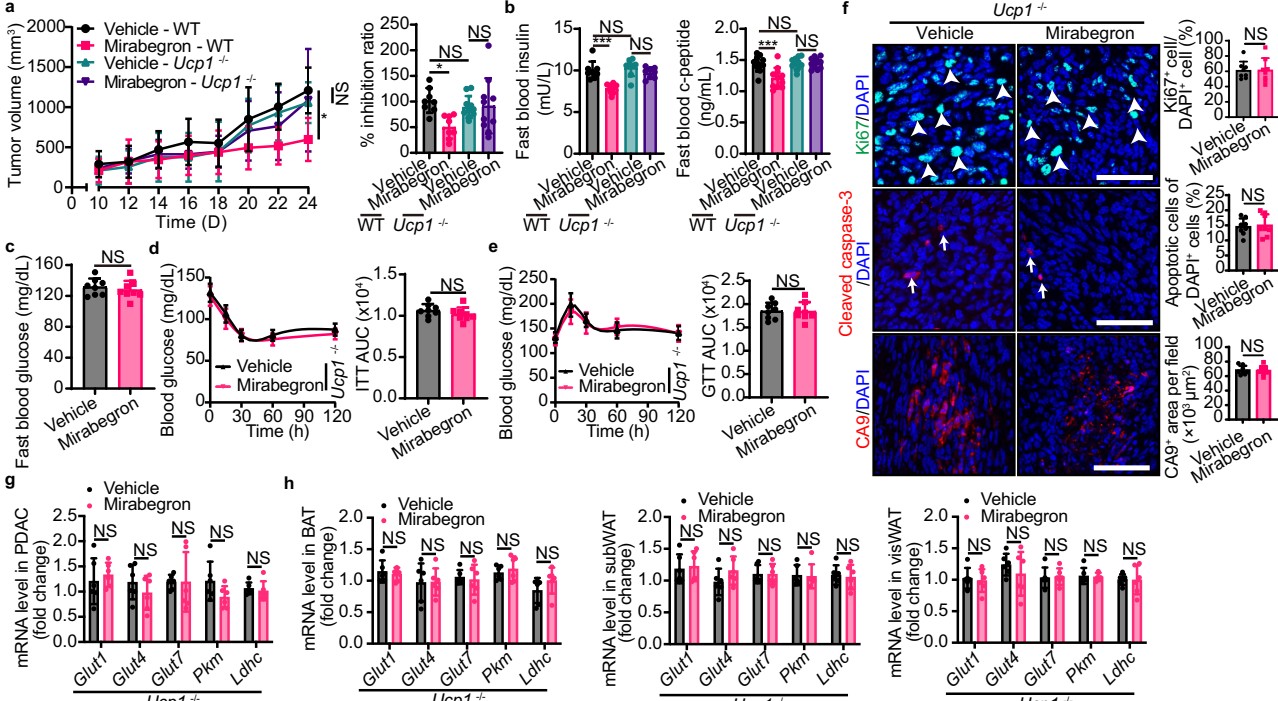

**Fig. 6 | Mirabegron-instigated tumor suppression is UCP1-dependent. a** Tumor growth, tumor inhibition ratio, and tumor weight of vehicle- and mirabegron-treated PDAC tumor-bearing WT and *Ucp1*[-/-] mice (*n* = 8 mice per group). **b** Fast blood insulin and c-peptide levels of vehicle- and mirabegron-treated PDAC tumor-bearing WT and *Ucp1*[-/-] mice (*n* = 8-11 mice per group). **c** Fast blood glucose levels of vehicle- and mirabegron-treated PDAC tumor-bearing *Ucp1*[-/-] mice (*n* = 8 mice per group). **d**, **e** Insulin tolerance test (ITT) and glucose tolerance test (GTT) of vehicle- and mirabegron-treated PDAC tumor-bearing *Ucp1*[-/-] mice (*n* = 8 mice per group). **f** Immunofluorescence staining of Ki67[+] proliferating cells (green), cleaved-caspase 3[+] apoptotic cells (red), and CA9[+] hypoxic area (red) of PDAC tumors. Tissues were counterstained with 4′,6-diamidino-2-phenylindole (DAPI, blue). Arrows and arrowheads point to their respective positive signals. Scale bar, 50 μm. Quantifications of Ki67[+] signal, cleaved-caspase 3[+] signal, and CA9[+] signal (*n* = 8 random fields per group). **g** qPCR analysis of GLUTs and glycolysis-related genes in vehicle- and mirabegron-treated PDAC tumors in *Ucp1*[-/-] mice (*n* = 6 samples per group). **h** qPCR analysis of GLUTs and glycolysis-related genes in BAT, subWAT, and visWAT in vehicle- and mirabegron-treated PDAC tumor-bearing *Ucp1*[-/-] mice (*n* = 6 samples per group). Statistical analysis was performed using one-way ANOVA test (**a**, **b**) and two-sided unpaired t-test (**c**, **h**). NS, not significant; *\**P* < 0.05; *\*\**P* < 0.01; *\*\*\**P* < 0.001. Data presented as mean ± SD. Each experiment was repeated at least three times and the representative experiment was shown (**a–h**). Source data are provided as a Source Data file.

glycolytic pathways employ inhibition of critical enzymes for glucose utilization and energy production[31,32]. To achieve this goal, these enzymes should be highly expressed in malignant cells but less or not expressed in proliferating healthy cells. To this end, antiglycolytic agents targeting GLUTs and glycolytic enzymes have been assessed for their anticancer effects in preclinical and clinical settings[33]. Despite their potent anticancer effects, inhibition of GLUTs that are ubiquitously expressed in most healthy cells has experienced considerable adverse effects. Numerous inhibitors of glycolytic enzymes have been developed and show anticancer effects in preclinical studies[4]. However, development of these antiglycolytic agents towards clinical use warrants efficacy assessments by designing rigorous clinical trials.

In contrast to directly targeting glycolytic pathways in cancer cells, activation of BAT and browning WAT targets non-tumor tissues in cancer hosts. Activation of thermogenic activity of BAT and browning WAT significantly alters glucose distribution between the tumor mass and adipose tissues. Compared with adipose depots, most clinically detectable tumor masses are relatively small in size. Since both thermogenic adipose tissues and tumor tissues utilize glucose for metabolism, activation of BAT/browning WAT would inevitably create a competitive situation for glucose uptake between tumor and fat tissues. Our [18]F-FDG-PET-CT data support the fact of minimal glucose uptake in tumors when BAT becomes activated by mirabegron. Marked reduction of glucose uptake in cancer cells by browning BAT/WAT would result in starvation of cancer cells by impeding the glycolysis-dependent mechanism. Indeed, high glucose feeding in tumor-bearing mice largely restored tumor growth under mirabegron

treatment. These findings demonstrate that impairment of glucose metabolism in cancer cells is the key mechanism of mirabegron-committed tumor suppression (Fig. S6).

A groundbreaking discovery in our present study is that genetic deletion of UCP1 in mice completely abrogates tumor suppression by mirabegron. These findings show that NST thermogenesis, but not just glucose uptake, in browning BAT/WAT is essential for tumor suppression. It is known that mirabegron-induced metabolic effects are mainly UCP1-dependent[22]. Of note, various UCP1-independent NST mechanisms such as creatine cycling[34], AMPK activation[35], SERCA2b-mediated calcium cycling[36], and ADP/ATP carrier[37] in adipose tissue have been reported. Whether activating these UCP1-independent NST mechanisms inhibits tumor growth requires further investigation. Other than glucose competition, NST-related metabolism could also produce metabolic products that may potentially participate in tumor suppression. Metabolism reduction in tumor cells might alleviate hypoxia, which may contribute to the tumor inhibitory effect. Systemic insulin alterations may contribute to the anti-tumor effect. In the present study, we cannot exclude these possibilities. Although surgical removal of BAT largely ablated tumor suppression by mirabegron, this operation alone is not sufficient to completely abrogate the anticancer effect, suggesting that other tissues also participate in tumor inhibition. In line with this hypothesis, removal of WAT also produces a significant albeit less anticancer neutralization effect, supporting browning WAT in tumor suppression. Our previously published work showed that mirabegron seems to augment a more robust browning phenotype than cold exposure[19]. Treatment of tumor-bearing mice

with mirabegron instigates browning of both subWAT and visWAT, which would enhance glucose uptake and metabolism in these fat depots. The extensive WAT browning by mirabegron treatment reveals the difference between mirabegron- and cold-induced tumor suppression, and explains that both BAT and WATs were involved in the mirabegron-suppressed tumor, whereas cold-inhibited tumor is only BAT-dependent[15].

In the present study, we use mirabegron as an example of the discovery of anticancer drugs. Of note, sympathetic activation has been reported to promote tumor growth through stress-induced β-adrenergic signaling. β-adrenergic signaling instigated tumor growth by directly upregulating signaling such as DNA damage repair and oncogenes in tumor cells, as well as by indirectly mobilizing glucose and fatty acids for fueling the tumor development. However, the majority of these studies have focused on β1 and β2 signaling. The antitumor effect of β3 has been largely overlooked by the field[38]. Besides mirabegron, we believe that other β3-AR agonists[39] or adipose browning approaches, including therapeutic agents[40], food[41], chemicals[42], devices[43], tissue transplantation[14], physical exercises[44], and food restriction[45] would also produce anticancer effects. Therefore, any approaches including medical intervention, chemical administration, or physical devices that could induce adipose tissue browning or activation of brown fat would be potentially used for cancer therapy. Thus, our discoveries have broad implications in cancer therapy and perhaps even in the treatment of non-malignant diseases.

Another important finding is that mirabegron is effective for the treatment of most aggressive and currently untreatable cancer types, such as PDAC and HCC. In preclinical cancer models, we show that mirabegron displays potent anticancer effects against these cancers. Since effective treatment of PDAC, HCC, and some other cancers remain an unmet medical need in the clinic, adipose browning drugs such as mirabegron would provide hopes for effective treatment of these otherwise untreatable cancers. This important point warrants clinical validation.

Taken together, in this study we present a distinct concept of cancer therapy by drugs that alter global metabolism in cancer hosts. This therapeutic paradigm employs mechanisms of NST in browning fats without directly targeting tumors. While almost all currently available anticancer drugs are aiming to target various cellular and molecular components in tumor tissues, we demonstrate that targeting non-tumor tissues such as adipose tissues in cancer hosts would be probably effective, less toxic, and a generalized approach for cancer therapy. In addition to fat tissues shown in this study, we anticipate that alteration of global metabolism in other tissues and organs, such as liver and muscle tissues, would also be effective for treating cancer.

## Methods

### Ethical regulations statement
All animal experiments were approved by the Animal Experimental Ethical Committee of Fudan University, Shanghai, China (20190430) and in accordance with the Animal Research: Reporting of In Vivo Experiments (ARRIVE) guidelines.

### Cell culture
Murine pancreatic cancer cell line Panc02 was kindly provided by Dr Maximilian Schnurr at the University of Munich, Germany[46]. Murine hepatocellular carcinoma cell line Hepa1-6 was purchased from ATCC (CRL-1830). Murine MC-38 colon adenocarcinoma cell line was kindly provided by Dr. Rubén Hernández at the University of Navarra, Spain. MC-38, Panc02, and Hepa1-6 cells were cultured and maintained in a Dulbecco's Modified Eagle's Medium (DMEM, D6429, Merck) supplemented with 10% FBS, 100 units/mL penicillin, and 100 µg/mL streptomycin (P4333, Merck)[47].

### Animals
Male 6-8-week-old C57BL/6 mice and male $Ucp1^{-/-}$ mice in the C57BL/6 background (Stock No. T037633) were purchased from GemPharmatech, China and maintained under a 12-hour dark/12-hour light cycle with food (SCXK2020-0004, Beijing Keao Xieli Feed) and water provided ad libitum. Male 9-10-week-old C57Bl/6 J background-$Apc^{Min/+}$ mice (Stock no. 002020) were obtained from the Jackson Laboratory. Mice were randomly divided into groups for all experiments. No statistical methods were used to predetermine sample size. The Investigators were not blinded to allocation during experiments and outcome assessment.

### Mouse tumor models
Approximately $1 \times 10^6$ MC-38 tumor cells and $1 \times 10^6$ Panc02 cells in 100 µL Phosphate buffered saline (PBS) were subcutaneously implanted into each C57BL/6 mouse. For the intraliver injection model, isoflurane-anesthetized 6-8-week-old male C57BL/6 mice were surgically operated on and were injected with approximately $0.5 \times 10^6$ Hepa1-6 cells in 40 µL PBS into the liver lobe, followed by a closure of incision with the sterile surgical suture (CR436, Jinhuan Medical, China). Tumor sizes were measured with a caliper and tumor volumes were calculated according to the standard formula (length × width$^2$ × 0.52)[48]. For the genetic spontaneous intestinal adenoma model, $Apc^{Min/+}$ mice, that usually developed multiple intestinal adenomas between weeks 10 and 14, were used. All tumor sizes were ≤ 2.5 cm$^3$ and did not exceed the maximal tumor size allowed by the ethical approval. Fresh tissues including tumors and adipose depots were harvested and kept in dry ice or liquid nitrogen for metabolomics analysis or RNA and protein extraction. A portion of fresh tissues was fixed with 4% paraformaldehyde (PFA) (MA0192, Meilunbio, China) overnight for histological and immunohistochemical analyses.

### Drug treatment
Mirabegron (223673-61-8, BOC sciences, USA) was dissolved in polyethylene glycol (PEG, 91893, Sigma-Aldrich) as a 100 mg/mL stock, which was further diluted in PBS upon use. Mirabegron was orally administrated in each mouse at 0.8, 3.2, 8, and 10 mg/kg bodyweight/day until mice were sacrificed for further analysis. CL-316, 243 (1499, Tocris Bioscience, USA) was injected into each mouse at 1 mg/kg bodyweight/day. Isoproterenol (HY-B0468, MedChemExpress, USA) was injected into each mouse at 1 mg/kg bodyweight/day. Propranolol (HY-B0573B, MedChemExpress, USA) was injected into each mouse at 2 mg/kg bodyweight/day. L-748,337 (HY-103211, MedChemExpress, USA) and SR59230A (HY-100672, MedChemExpress, USA) were injected into each mouse at 5 mg/kg bodyweight/day. Drugs were given to tumor-bearing mice after 14 days of tumor implantation when tumor size reached approximately 0.2 cm$^3$.

### Climate chamber
For room temperature and thermoneutrality exposure, mice were housed in a climate chamber (MGC-400H, Yiheng, China) an at the Department of Cellular and Genetic Medicine, School of Basic Medical Sciences, Fudan University. Mice were implanted with tumor and exposed to 22 °C or thermoneutral temperature of 30 °C.

### Adipose depot removal
For BAT removal, a small incision was surgically created at the intrascapular area under anaesthetization. Blood vessels were pinched to prevent excessive bleeding. BAT was carefully dissected, followed by the suturing of the incision. For subWAT and visWAT removal, mice were kept in the supine position and an abdominal incision was surgically created. WAT depots were blunt dissected and removed by autoclaved surgical scissors, followed by the suturing of the peritoneum and abdominal skin. After adipose depot removal, mice were closely observed until full recovery for 1 week before further treatments.

## Survival assay

Survival studies of tumor-bearing mice were performed at Fudan University, Shanghai, China according to the ethical permit in which the humane endpoint (tumor size ≤ 3 cm³ or body condition score (BCS) ≤ 1) was used to sacrifice each mouse. This tumor size or BCS-1 (moribund state) was not reached in any of the experimental mice.

## PET–CT imaging

After 2 weeks of vehicle- or mirabegron-treatment, tumor-bearing mice were fasted for 6 h before PET–CT scanning. ¹⁸F-FDG was produced by a cyclotron (Siemens CTI RDS Eclips ST) using the Explora FDG4 module. In vivo PET–CT imaging scans and image analysis were performed using an Inveon Animal-PET-CT system (Siemens Preclinical Solution). Mice were scanned 1 h after intravenous injection of 3.7 MBq (100 µCi) of ¹⁸F-FDG. Experimental animals were maintained with anaesthetization using 2.5% isoflurane/oxygen before and during the scanning. Three-dimensional ordered-subset expectation maximization (3D-OSEM)/maximum algorithm was used for image reconstruction. The maximal percentage-injected dose per gram was calculated and the SUV-BW of BAT and tumor was measured in the region of interest. Inveon Acquisition Workplace software (Siemens Medical Solutions) was used for further analysis.

## Blood glucose, insulin, c-peptide, insulin tolerance, and glucose tolerance tests

Mice were starved for 4-6 h before tests. For the blood glucose level test, blood samples from tail vein were collected and measured by a glucometer (Accu-Chek, Active, Roche Diagnostics).

For the blood insulin and c-peptide level test, blood samples from retroorbital venous plexus were collected and measured by the insulin elisa kit (EMINS, Invitrogen, USA) and c-peptide test kit (90050, Crystalchem, USA). For the insulin tolerance test, blood samples from tail vein were collected and measured by a glucometer (Accu-Chek, Active, Roche Diagnostics) immediately before and at 15, 30, 60 and 120 min after intraperitoneal injection of insulin (0.5 U per kg bodyweight). For glucose tolerance test, glucose levels were tested before and at 15, 30, 60 and 120 min after oral feeding with 1.5 mg glucose with 10 µL per g bodyweight.

## Metabolic analyses

Whole-body energy metabolism was measured by oxygen consumption using an open-circuit system (Sable, USA). Tumor-bearing mice with tumor size of approximately 1.0 cm³ were kept in the measuring enclosure system for 5 h to adapt to the environment and reduce environmental stress. Prior to measurement, mice were anesthetized by i.p. injection with pentobarbital at the dose of 90 mg/kg. Animals were immediately transferred to a 30 °C metabolic chamber, where $O_2$ consumption and release of $CO_2$ were uninterruptedly recorded for 30 min.

## Metabolomics

Metabolites were extracted from snap-frozen tumor samples using a 400 µL methanol: water (4:1, v/v) solution in a high throughput tissue crusher (Wonbio-96C, Shanghai Wanbo Biotechnology, China) at 50 Hz for 6 min, then followed by vortex for 30 s and ultrasound at 40 kHz for 30 min at 5 °C. The samples were placed at −20 °C for 30 min, followed by centrifugation at 13000 g at 4 °C for 15 min. The supernatant was transferred to sample vials for LC-MS/MS analysis. Pooled quality control samples were prepared and tested in the same manner as the analytic samples to monitor the stability of the analysis. Chromatographic separation of the metabolites was performed on a Thermo UHPLC system equipped with an ACQUITY BEH C18 column (100 mm × 2.1 mm i.d., 1.7 µm; Waters, Milford, USA). The mobile phases consisted of 0.1% formic acid in water (solvent A) and 0.1% formic acid in acetonitrile: isopropanol (1:1, v/v)(solvent B). The solvent gradient changed according to the following conditions: from 0

to 3 min, 95% (A): 5% (B) to 80% (A): 20% (B); from 3 to 9 min, 80% (A): 20% (B) to 5% (A): 95% (B); from 9 to 13 min, 5% (A): 95% (B) to 5% (A): 95% (B); from 13 to 13.1 min, 5% (A): 95% (B) to 95% (A): 5% (B), from 13.1 to 16 min, 95% (A): 5% (B) to 95% (A): 5% (B) for equilibrating the systems. The sample injection volume was 2 µL and the flow rate was set to 0.4 mL/min. The column temperature was maintained at 40 °C. The mass spectrometric data was collected using a Thermo UHPLC-Q Exactive Mass Spectrometer equipped with an electrospray ionization (ESI) source. The optimal conditions were set as followed: Aus gas heater temperature, 400 °C; sheath gas flow rate, 40 psi; Aus gas flow rate, 30 psi; ion-spray voltage floating (ISVF), −2800V in negative mode and 3500 V in positive mode; normalized collision energy, 20-40-60 V rolling for MS/MS. Data acquisition was performed with the Data Dependent Acquisition (DDA) mode. The detection was carried out over a mass range of 70-1050 m/z.

After UPLC-TOF/MS analyses, the raw data were imported into the Progenesis QI 2.3 (Nonlinear Dynamics, Waters, USA) for peak detection and alignment. Metabolic features detected at least 80% in any set of samples were retained. Mass spectra of these metabolic features were identified by databases including Human metabolome database (HMDB) (http://www.hmdb.ca/) and Metlin database (https://metlin.scripps.edu/). For metabolites having MS/MS confirmation, only the ones with MS/MS fragments score above 30 were considered as confidently identified. Statistical analysis was performed using ropls (Version1.6.2, http://bioconductor.org/packages/release/bioc/html/ropls.html) R package from Bioconductor on Majorbio Cloud Platform (https://cloud.majorbio.com). Unsupervised principal component analysis (PCA), orthogonal partial least squares discriminate analysis (OPLS-DA), variable importance in the projection (VIP), and differential metabolites analysis were performed. Metabolites were further mapped into their biochemical pathways through metabolic enrichment and pathway analysis based on the KEGG database (http://www .genome.jp/kegg/).

## Histology and immunohistochemistry

The paraffin-embedded tissue sections of 5-µm thickness were incubated at 60 °C for 2 h, deparaffinized in Xylene (Cat. No. 10023418, SCR, China), and sequentially rehydrated in 99%, 95%, and 70% ethanol (Cat. No. 10009218, SCR, China). Tissue slides were counterstained with Haematoxylin (Mayer's) (Cat. No. MB9897, Meilunbio, China) and Eosin (Cat. No. MA0164, Meilunbio, China). Stained tissues were analyzed under a light microscope (Leica DM IL LED). For immunohistochemistry, tissue slides were stained with a rabbit anti-mouse cleaved caspase-3 antibody (1:200, 9661, Cell Signaling), a rabbit anti-mouse Ki67 antibody (1:100, PA5-19462, Thermo Fisher Scientific), a rabbit anti-CA9 antibody (1:300, NB100-417, NOVUS), a rabbit anti-mouse UCP1 antibody (1:200, ab10983, Abcam), a rabbit anti-mouse COX4 antibody (1:300, GTX114330, GeneTex), or a guinea pig anti-mouse Perilipin antibody (1:300, 20R-PP004, Fitzgerald Industries), followed by staining with species-matched secondary antibodies as follows: an Alexa Fluor 555-labeled goat anti-rabbit antibody (1:300, A21482, Thermo Fisher Scientific), an Alexa Fluor 488-labeled goat anti-rabbit antibody (1:300, A11008, Thermo Fisher Scientific), or an Alexa Fluor 647-labeled goat anti-guinea pig antibody (1:200, A-21450, Thermo Fisher Scientific). Positive signals were captured using a fluorescence microscope (Olympus BX53, Japan) and were further analyzed using an Adobe Photoshop CS software program.

## RNA isolation and PCR analysis

Total RNA was extracted from tissues using TRIeasy Total RNA Extraction Reagent (10606ES60, YEASEN, China) and was reversely transcribed using Hifair II 1st Strand cDNA Synthesis SuperMix (Cat. No. 11123ES60, YEASEN, China). Reverse transcription was performed at 42 °C for 30 min, subsequently 85 °C for 5 min to inactivate the enzyme activity. The cDNA samples were stored at −20 °C and

subjected to qPCR using an ABI Prism 7500 System (Applied Biosystems). Each sample was triplicated and in a 20 µL reaction containing Hieff qPCR SYBR Green Master Mix (Cat. No. 11203ES03, YEASEN, China), 200 nM forward and reverse primers, and 1 µL cDNA. The qPCR protocol was executed for 40 cycles and each cycle consisted of hold at 95 °C for 10 s, denaturation at 95 °C for 15 s, annealing at 60 °C for 1 min, and extension at 72 °C for 1 min. The primer pairs specific for various genes used in our experiments included: mouse *Gapdh* forward: 5′-CCAGCAAGGACACTGAGCAA-3′; mouse *Gapdh* reverse: 5′-GG GATGGAAATTGTGAGGGA-3′; mouse *Glut1* forward: 5′-GCAGTTCG GCTATAACACTGG-3′; mouse *Glut1* reverse: 5′-GCGGTGGTTCCATG TTTGATTG-3′; mouse *Glut4* forward: 5′-ACACTGGTCCTAGCTG-TATTCT-3′; mouse *Glut4* reverse: 5′-CCAGCCACGTTGCATTGTA-3′; mouse *Glut7* forward: 5′-GGACAAAGAGATAGGGACACCT-3′; mouse *Glut7* reverse: 5′-GCAGCACTTAGGGTAGTCAGG-3′; mouse *Pkm* forward: 5′-CGCCTGGACATTGACTCTG-3′; mouse *Pkm* reverse: 5′-GA AATTCAGCCGAGCCACATT-3′; mouse *Ldhc* forward: 5′-GGAAGATAAA CTTTCCCGGTGT-3′; mouse *Ldhc* reverse: 5′-TTCATCAGCCAAACCCT TCAG-3′; mouse *Ehhadh* forward: 5′- ATGGCTGAGTATCTGAGGCTG-3′; mouse *Ehhadh* reverse: 5′- ACCGTATGGTCCAAACTAGCTT-3′; mouse *Acox2* forward: 5′- CACCCTGACATAGACAGTGAAAG-3′; mouse *Acox2* reverse: 5′- CTGGGTCACGTTGGATGAGG-3′; mouse *Acot12* forward: 5′- GCTGCTCAAGTGGATGGACA-3′; mouse *Acot12* reverse: 5′- CTTGTGCT GAACGCCCTAGTC-3′; mouse *Acad11* forward: 5′- TGACACCGTGGA AGTGCTAC-3′; mouse *Acad11* reverse: 5′- CCCGGCAAGTGCTGATTCA-3′; mouse *Acsm3* forward: 5′- CTTTGGCCCCAGCAGTAGATG-3′; mouse *Acsm3* reverse: 5′- GGCTGTCACTGGCATATTTCAT-3′; mouse *Acsl1* forward: 5′- TGCCAGAGCTGATTGACATTC-3′; mouse *Acsl1* reverse: 5′- GGCATACCAGAAGGTGGTGAG-3′; mouse *Acox1* forward: 5′- TAACTTC CTCACTCGAAGCCA-3′; mouse *Acox1* reverse: 5′- AGTTCCATGACC-CATCTCTGTC-3′; mouse *Acsm5* forward: 5′- GCTTCCGAGACTCCCA-GATTG-3′; mouse *Acsm5* reverse: 5′- CCGATACTTGAGATCCTTCGC-3′; mouse *Crot* forward: 5′- AGAACGGACATTTCAGTACCAGG-3′; mouse *Crot* reverse: 5′- TTCCAGCCAGTTTCGTTTTCC-3′; mouse *Fabp1* forward: 5′- ATGAACTTCTCCGGCAAGTACC-3′; mouse *Fabp1* reverse: 5′-GGTCCTCGGGCAGACCTAT-3′; mouse *Slc27a2* forward: 5′- CGAGACGA GACGCTCACCTA-3′; mouse *Slc27a2* reverse: 5′- ACGAATGTTGTAGT TGAGGCAC-3′; mouse *Slc27a5* forward: 5′- TCTATGGCCTAAAGTT-CAGGCG-3′; mouse *Slc27a5* reverse: 5′- CTTGCCGCTCTAAAGCATCC-3′; mouse *Cd36* forward: 5′- ATGGGCTGTGATCGGAACTG-3′; mouse *Cd36* reverse: 5′- TTTGCCACGTCATCTGGGTTT-3′; mouse *Fabp4* forward: 5′- AAGGTGAAGAGCATCATAACCCT-3′; mouse *Fabp4* reverse: 5′- AAGGTGAAGAGCATCATAACCCT-3′; mouse *Fabp5* forward: 5′- AAAGA GCTAGGAGTAGGACTGG-3′; mouse *Fabp5* reverse: 5′- TGTTGCCATCA CACGTAATGA-3′; mouse *Cpt1a* forward: 5′- TGGCATCATCACTGGTG TGTT-3′; mouse *Cpt1a* reverse: 5′- GTCTAGGGTCCGATTGATCTTTG-3′; mouse *Cpt1c* forward: 5′- TCTTCACTGAGTTCCGATGGG-3′; mouse *Cpt1c* reverse: 5′- ACGCCAGAGATGCCTTTTCC-3′; mouse *Cpt2* forward: 5′- CAGCACAGCATCGTACCCA-3′; mouse *Cpt2* reverse: 5′- TCCCAATG CCGTTCTCAAAAT-3′; mouse *Slc27a4* forward: 5′- ACTGTTCTCCAAGC TAGTGCT-3′; mouse *Slc27a4* reverse: 5′- GATGAAGACCCGGATGAAAC G-3′; mouse *Pgm2* forward: 5′- CAGAACCCTTTAACCTCTGAGTC-3′; mouse *Pgm2* reverse: 5′- CGAGAAATCCCTGCTCCCATAG-3′; mouse *Pgam2* forward: 5′- ACCACCCACCGCCTAGTAAT-3′; mouse *Pgam2* reverse: 5′- GCACCCACATTTGGTCCGTAA-3′; mouse *Fabp5* forward: 5′- AAAGAGCTAGGAGTAGGACTGG-3′; mouse *Fabp5* reverse: 5′- TGTTG CCATCACACGTAATGA-3′; mouse *Hk2* forward: 5′- ATGATCGCCTGC TTATTCACG-3′; mouse *Hk2* reverse: 5′- CGCCTAGAAATCTCCAGA AGGG-3′; mouse *Pgam1* forward: 5′- AGCGACACTATGGCGGTCT-3′; mouse *Pgam1* reverse: 5′- TGGGACATCATAAGATCGTCTCC-3′; mouse *Pfkp* forward: 5′- CGCCTATCCGAAGTACCTGGA-3′; mouse *Pfkp* reverse: 5′- CCCCGTGTAGATTCCCATGC-3′; mouse *Hk1* forward: 5′- AA CGGCCTCCGTCAAGATG-3′; mouse *Hk1* reverse: 5′- GCCGAGATCCAGT GCAATG-3′; mouse *Pfkl* forward: 5′- GGAGGCGAGAACATCAAGCC-3′; mouse *Pfkl* reverse: 5′- GCACTGCCAATAATGGTGCC-3′; mouse *Hk3*

forward: 5′- TGCTGCCCACATACGTGAG-3′; mouse *Hk3* reverse: 5′- GCC TGTCAGTGTTACCCACAA-3′;

## Statistics & Reproducibility
Statistical computations were performed using GraphPad Prism (GraphPad, USA). Statistical differences between two groups were determined by a two-tailed Student's t test. $P < 0.05$ was considered statistically significant, $P < 0.01$ was very significant, and $P < 0.001$ was extremely significant. Statistical differences among multiple groups were evaluated using a one-way ANOVA test. The variance was similar between the groups that were being statistically compared. No statistical methods were used to predetermine the sample size. No data were excluded from the analyses. Age, gender, and background-matched mice were randomly allocated into the groups for all animal experiments. The Investigators were not blinded to allocation during experiments and outcome assessment. The data is presented as means ± standard deviation (SD).

## Reporting summary
Further information on research design is available in the Nature Portfolio Reporting Summary linked to this article.

## Data availability
The metabolomics raw data generated in this study have been deposited in the MetaboLights database under accession code MTBLS8682. All data associated with this study are present in the paper or Supplementary Information. All materials are commercially available and the supplier names and catalog numbers have been provided in Methods. All resources generated in this study are available upon request from the corresponding author. Source data are provided with this paper.

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

## Acknowledgements

We thank Dr. Jianping Zhang and Dr. Qi Sun in Fudan University, Shanghai, China for their technical support. We thank Dr. Rui Hao, Dr. Fahuan Song, and Dr. Yiwen Zhang in Zhejiang Provincial People's Hospital, Hangzhou, China for their technical support. We thank Ms. Yu Wang in Shanghai University of Traditional Chinese Medicine for her artistic input in schematic diagrams. Y.Y. is supported by the National Program on Key Research (2021YFA0804700), the National Youth Talent Support Program (Ten Thousand Talent Program), the Innovation Research Team of High-level Local Universities in Shanghai, and the Program for Professor of Special Appointment in Shanghai (Eastern Scholar, project no. TP2018007). X.S. is supported by National Natural Science Foundation of China (Project No. 82204857). T.S. is supported by the Scandinavia-Japan Sasakawa Foundation (GA23-JP-0040). Y.C.'s laboratory is supported through research grants from the Swedish Research Council (project no. 2016-02215, 2019-01502, 2020-06121 and 2021-06122), National Key R&D Program of China (project no. 2020YFC0846600), the Hong Kong Centre for Cerebro-cardiovascular Health Engineering, the Swedish Cancer Foundation (project no. 200734PjF), the Swedish Children's Cancer Foundation (project no. PR2018-0107), the Strategic Research Areas (SFO)–Stem Cell and Regenerative Medicine Foundation, the Karolinska Institute Foundation (project no. 2020-02080), the Karolinska Institute distinguished professor award, the Karolinska Institute Foundation (project no. 2020-02588).

## Author contributions

Y.C. generated the ideas and designed experiments. X.S., W.S., Y.Y., Z.M., S.X., J.D. and S.L. performed most experiments and organized all figures. Y.W., P.H., X.L. and M.G. provided important materials or reagents. T.S., J.W., X.J., X.H., Y.W., P.H., X.L. and M.G. participated in discussions. Y.Y. and Y.C. wrote the manuscript.

## Funding

## Competing interests

All authors claim no conflict of interest related to this work.
