## [Peer Review File · Nature Communications]

Mirabegron displays anticancer effects by globally browning adipose tissuesEditorial Note: This manuscript has been previously reviewed at another journal that is not operating a transparent peer review scheme. This document only contains reviewer comments and rebuttal letters for versions considered at *Nature Communications*.

REVIEWER COMMENTS

Reviewer #1 (Remarks to the Author):

In Sun et al.'s "Mirabegron signifies a new paradigm of anticancer drug therapy by globally browning adipose tissues," the authors report that the β 3-adrenergic receptor (AR) agonist mirabegron has anti-tumor effects mediated through the browning of adipose tissue and the associated increase in UCP1-mediated thermogenesis. They conclude that their "findings demonstrate that mirabegron represents a novel class of anticancer drugs with a new mechanistic paradigm for the effective treatment of various cancer types."

These findings represent a potential major change in the understanding of the interaction between adrenergic activation and tumor growth. Given this importance, several additional studies would help improve the authors' model.

The manuscript could be improved by addressing the following comments:

A. Uniqueness of mirabegron and the use of controls

The authors describe mirabegron as part of a "novel class of anticancer drugs," but the class itself is not novel. Members of its class have been used in animal studies for three decades. It is good to see that the authors examined another β 3-AR agonist, CL-316,243, which in Fig. S3A shows some inhibition of PDAC tumor growth. However, mirabegron and CL-316,243 were not compared, and no general or specific inhibitors of mirabegron/CL were used to demonstrate specificity or a dose-response.

Beyond CL-316,243, norepinephrine is the endogenous neurotransmitter and hormone, and it is the actual physiological activator of BAT thermogenesis. Isoproterenol is a non-specific β -AR agonist that matches nearly all of the effects of mirabegron, and then some. Meanwhile, propranolol is a beta-AR antagonist that blocks mirabegron's effects and is useful in establishing its pharmacology. L-748,328 and L-748,337 are β 3-AR specific and can help determine which adrenergic receptors are mediating anti-tumor activity.

The authors need to repeat several of their experiments with proper positive and negative controls to show whether mirabegron alone has anti-tumor activity, or if the effect is a class effect seen in beta-adrenergic agonists. Examples of helpful studies:

1. Head-to-head-to-head comparisons of mirabegron, CL-316,243, norepinephrine, and isoproterenol, in vitro and in vivo.
2. Dose-response studies of mirabegron's effects.
3. Inhibitor studies using mirabegron and propranolol and also L-748,328 or L-748,337 as specific and non-specific β 3-AR inhibitors, respectively.

B. Mechanism

The authors propose that “mirabegron-induced NST results in the downregulation of multiple metabolic pathways in tumors.” A diagram is shown in Fig. S6. Potential mechanisms they offer include the following:

- a. Reduced glucose supply in tumors;
- b. Trimmed glucose transportation in tumor cells;
- c. Reduced glucose metabolism in tumor cells;
- d. Alleviation of tumor hypoxia;
- e. Downregulated lipid metabolism in tumor cells

While some of these mechanisms can be tested in future experiments, some have already been done in this study. Figs 2C-E show that blood glucose in the mirabegron-treated mice is lower, but certainly within the normal range. Do the authors believe that a normal fasting glucose is too low for tumors to grow?

Option (e) could be tested by measuring blood lipids. If they are unchanged, then (d) is the only remaining mechanism, the alleviation of hypoxia; the authors should then explain how mirabegron treatment would lead to this.

C. BAT thermogenesis

C1. Fig. S6 argues that UCP1 deficiency abrogates mirabegron-induced adipose tissue browning. It is not clear why this would be as UCP1 is downstream and part of a separate pathway initiated by adrenergic signaling. By what mechanism is this happening?

C2. Kazak and Spiegelman have published numerous studies showing that UCP1 is only one of several futile cycles that leads to thermogenesis in BAT. One notable alternative is futile creatine cycling. Meanwhile, UCP1-KO leads to disrupted mitochondrial structure and function beyond uncoupling and thermogenesis. The authors need additional studies to show that a loss of uncoupling/futile cycles is what leads to the reported anti-tumor effects.

Reviewer #2 (Remarks to the Author):

The authors addressed all of my concerns convincingly

Reviewer #3 (Remarks to the Author):

With their revisions, Sun et al have been responsive to the previous round of reviews, significantly increasing the "interpretability" of the manuscript by updating many of the figures to provide actual levels of growth factors. Despite these improvements having removed the human data from this version of the manuscript it is difficult to say if the current version can support the authors largest claims as written.

Figure 1 clearly shows that with Mirabegron treatment that several tumor models show decreased growth rates (not as stated inhibition, just decreased growth rates).

Figure 2 demonstrates that Mirabegron treatment increases UPC1 expression in adipose tissues, along with a decrease in adipocyte cell size in WAT and BAT. They show that treatment decreases blood glucose and insulin levels while increasing metabolic rate as gauged by O₂ consumption.

Figure 3 they show that if they remove the BAT that they significantly reduce the impact of Mirabegron treatment upon tumor growth, their markers of choice (Ki67, CA9, CC3), and glucose. Despite requests in the last round of review they did not address the changes in insulin nor c-peptide in these models.

Figure 4 they argue that Mirabegron treatment inhibits glucose metabolism in the tumor tissue, but they do not control for the impact of insulin/glucose changes in this data set as (again) insulin and glucose levels are associated with the changes they demonstrate in glucose regulating enzymes in the tumor. This feature is not controlled for in their experimental design. This is a key feature since they go on to show that exogenous glucose reduces the impact of Mirabegron in figure 5. Similarly they show that UCP1 deletion abrogates the impact of Mirabegron in figure 6.

For both figures 5 and 6 the pre-requisite matched controls are not being run in the same experiment. E.g. in figure 5 we only see mice with 15%HG and no experimental control for the effects is shown within the experiment; similarly there is no syngeneic control for the UCP1 experiments in figure 6. This makes it difficult to gauge the impact of their perturbations as shown/argued.

Manuscript NCOMMS-23-13681-T: Responses to reviewers' comments

Our point-by-point responses to the reviewers' comments are as follows:

Reviewer Comments

Reviewer: 1

Comment: In Sun et al.'s "Mirabegron signifies a new paradigm of anticancer drug therapy by globally browning adipose tissues," the authors report that the β 3-adrenregic receptor (AR) agonist mirabegron has anti-tumor effects mediated through the browning of adipose tissue and the associated increase in UCP1-mediated thermogenesis. They conclude that their "findings demonstrate that mirabegron represents a novel class of anticancer drugs with a new mechanistic paradigm for the effective treatment of various cancer types."

These findings represent a potential major change in the understanding of the interaction between adrenergic activation and tumor growth. Given this importance, several additional studies would help improve the authors' model.

Response: We thank the reviewer for these constructive comments and accurate understanding of our findings. We particularly appreciate the reviewer's comments about the potential major impact of our work on understanding the interactions between adrenergic activation and tumor growth.

Comment: A. Uniqueness of mirabegron and the use of controls

The authors describe mirabegron as part of a "novel class of anticancer drugs," but the class itself is not novel. Members of its class have been used in animal studies for three decades. It is good to see that the authors examined another β 3-AR agonist, CL-316,243, which in Fig. S3A shows some inhibition of PDAC tumor growth. However, mirabegron and CL-316,243 were not compared, and no general or specific inhibitors of mirabegron/CL were used to demonstrate specificity or a dose-response.

Response: We thank the reviewer for this important comment. We completely agree with the reviewer that mirabegron, which is clinically used for treating overactive bladder diseases, as a class of drug per se is not novel. In this work, we originally meant to say it is a novel anticancer drug. In order to avoid any confusions for readers, we have revised the text to remove the "new class". On the basis of the reviewer's recommendation, we have 1) side-by-side compared the anticancer effect of mirabegron with CL-316,243; and 2) used general or specific inhibitors of mirabegron/CL in our experimental settings. These new experimental data are described below.

Comment: Beyond CL-316,243, norepinephrine is the endogenous neurotransmitter and hormone, and it is the actual physiological activator of BAT thermogenesis. Isoproterenol is a

non-specific β -AR agonist that matches nearly all of the effects of mirabegron, and then some. Meanwhile, propranolol is a beta-AR antagonist that blocks mirabegron's effects and is useful in establishing its pharmacology. L-748,328 and L-748,337 are β_3 -AR specific and can help determine which adrenergic receptors are mediating anti-tumor activity.

The authors need to repeat several of their experiments with proper positive and negative controls to show whether mirabegron alone has anti-tumor activity, or if the effect is a class effect seen in beta-adrenergic agonists. Examples of helpful studies:

1. Head-to-head-to-head comparisons of mirabegron, CL-316,243, norepinephrine, and isoproterenol, in vitro and in vivo.

Response: We highly appreciate the reviewer for his/her authoritative advice, which is extremely helpful for us to understand the anticancer effect of mirabegron and AR agonists.

On the basis of the reviewer's suggestion, we have performed new experiments to allow us to head-to-head compare anticancer effects of mirabegron, CL-316,243, and isoproterenol, in vitro and in vivo. In the in vitro experiments, mirabegron, CL-316,243 and isoproterenol had no effect on tumor cell growth (See below Fig. R1). These data show that these adrenergic receptor activators have no direct impact on tumor cell proliferation in vitro. These new results have been incorporated into Fig. S3 of the revised manuscript.

R1

A

Fig. R1.

A. Tumor cell growth rates of Mirabegron-, CL-316,243- and isoproterenol-treated mc38 cells (n = 6 samples per group).

In the in vivo experiments, mirabegron, CL-316,243 and isoproterenol significantly inhibited tumor growth, indicating the activation of the sympathetic system by these β_3 -AR is required (See below Fig. R2). These new results have been incorporated into Fig. S3 of the revised manuscript.

R2

Fig. R2.

A. Tumor growth and tumor inhibition ratios of Vehicle-, Mirabegron-, CL-316,243- and isoproterenol-treated CRC (n = 8 mice per group).

Comment: 2. Dose-response studies of mirabegron's effects.

Response: On the basis of the reviewer's suggestion, we have performed new experiments to study the dose-dependent effect of mirabegron on tumor suppression. In the CRC tumor-bearing mice, the dose of 0.8 mg/kg daily lacked a statistically significant anticancer effect. At the dose of 8 mg/kg daily, mirabegron reached the maximal anticancer effect, and further increasing the dosage to 10 mg/kg daily did not enhance the anticancer effect beyond that of 8 mg/kg daily (Fig. R3). These new results have been incorporated into Fig. S1 of the revised manuscript.

R3

Fig. R3.

A. Tumor growth and tumor inhibition ratios of Vehicle- and 0.8, 3.2, 8, 10mg/kg Mirabegron-treated CRC (n = 8 mice per group).

Comment: 3. Inhibitor studies using mirabegron and propranolol and also L-748,328 or L-748,337 as specific and non-specific β 3-AR inhibitors, respectively.

Response: Agree. We have now performed new experiments using various β -AR antagonists in in vivo tumor models. We should emphasize that L-748,328 is not available for us to perform the experiments. We have another β 3-AR specific inhibitor, SR59230A, which has been reported as a specific inhibitor for β 3-AR (PMID: 8569714). Thus, we have used propranolol, L-748,337, and SR59230A for our in vivo tumor studies. Propranolol, L-748,337, and SR59230A significantly neutralized the antitumor effect of mirabegron. The effects of mirabegron alone, mirabegron plus propranolol, mirabegron plus L-748,337, and mirabegron plus SR59230A on tumor growth are statistically insignificant (Fig.R4). These new results have been incorporated into Fig. S3 of the revised manuscript.

R4

Fig. R4.

A. Tumor growth and tumor inhibition ratios of Vehicle- and Mirabegron-treated CRC with or without propranolol, L-748,337, and SR59230A (n = 7-8 mice per group).

Comment: B. Mechanism

The authors propose that “mirabegron-induced NST results in the downregulation of multiple metabolic pathways in tumors.” A diagram is shown in Fig. S6. Potential mechanisms they offer include the following:

- Reduced glucose supply in tumors;
- Trimmed glucose transportation in tumor cells;
- Reduced glucose metabolism in tumor cells;
- Alleviation of tumor hypoxia;
- Downregulated lipid metabolism in tumor cells

While some of these mechanisms can be tested in future experiments, some have already been done in this study. Figs 2C-E show that blood glucose in the mirabegron-treated mice is lower,

but certainly within the normal range. Do the authors believe that a normal fasting glucose is too low for tumors to grow?

Option (e) could be tested by measuring blood lipids. If they are unchanged, then (d) is the only remaining mechanism, the alleviation of hypoxia; the authors should then explain how mirabegron treatment would lead to this.

Response: We thank the reviewer for this important comment. In general, we believe that all listed mechanisms may play a role in mirabegron-inhibited tumor growth, although further studies are needed to provide supportive evidence.

Using ^{18}F -FDG, a glucose analogue, we have shown that activated BAT significantly increased glucose uptake and usage, whereas tumor decreased glucose utilization (Fig. 2B). Metabolomics and RNA-seq further confirmed the decreased utilization of glucose in solid tumors. As the reviewer pointed out, the reduced blood glucose levels are within the normal range, and we agree that blood glucose reduction alone may not be sufficient to inhibit tumor growth. Broader mechanisms may be involved. For example, downregulation of GLUT1 as the key limiting step of glucose utilization in tumor cells (Fig. R5). It is highly plausible that downregulation of GLUT1 is the key mechanism for the impairment of the glycolytic pathway in tumors, even though the blood glucose levels are within normal range. Therefore, complex mechanisms are likely to be involved in impaired tumor growth.

R5

Fig.R5.

A. mRNA level of the glucose transporter-related genes in vehicle- and mirabegron-treated mice (n = 6 samples per group).

Additionally, central carbon metabolism-related enzymes including phosphoglucomutases (Pgm2), phosphoglycerate mutase (Pgam1, Pgam2), hexokinases (Hk1, Hk2, Hk3), pyruvate kinase M1/2 (Pkm), and phosphofructokinases (Pfkp, Pfk1) were significantly inhibited (Fig. R6), indicating further impairment of tumor glycolysis. We performed a targeted metabolomic analysis of the tumor tissue glycolysis. Several crucial components of the glycolytic pathway, including glucose 1-phosphate, glucose 6-phosphate, fructose 6-phosphate, fructose 1,6-bisphosphate, 3-phosphoglyceric acid, phosphoenolpyruvic acid, glycerol 3-phosphate, gluconic acid, gluconic acid, 6-phosphogluconic acid, pyruvic acid, isocitric acid, citric acid, cis-aconitic

acid, and succinic acid, were markedly decreased in tumor tissues after mirabegron treatment (Fig. R6). These new results have been incorporated into Fig. 4 of the revised manuscript.

Fig.R6.

A. Heatmaps of the glycolysis-related genes (n = 6 samples per group).

B. Heatmaps of the glycolysis-related metabolites (n = 3 samples per group).

Regarding the alteration of lipid metabolism, we have performed a new analysis of the metabolic pathways for non-glycolytic energy supplies. Reduction patterns were evident in metabolites such as acyl-CoA and enzymes related to lipid metabolism (Fig. R7). This would support the possible mechanism (e). These new results have been incorporated into Fig. S5 of the revised manuscript.

Fig.R7.

A. Heatmaps of the different Acyl-CoAs (n = 3 samples per group).

B. Heatmaps of the FAO-related genes (n = 6 samples per group).

C. Heatmaps of the FFA transporter-related genes (n = 6 samples per group).

Concerning the alleviation of tumor hypoxia by β 3-AR activation, there are several possible mechanisms, including 1) delayed tumor growth rate; and 2) reduced accumulation of metabolites owing to impaired glycolysis. This issue has now been discussed in the revised manuscript.

Comment: C. BAT thermogenesis

C1. Fig. S6 argues that UCP1 deficiency abrogates mirabegron-induced adipose tissue browning. It is not clear why this would be as UCP1 is downstream and part of a separate pathway initiated by adrenergic signaling. By what mechanism is this happening?

Response: This is an excellent question. At the time of this writing, we have no good explanation for this excellent question. However, we speculate that there might be a feedback mechanism between UCP1 activation and the browning phenotype. One of the possibilities is that genetic deletion of UCP1 disrupts mitochondrial structure, which is the key determinant for adipose browning. It is possible that the disrupted mitochondria show reduced expression of other mitochondrial markers, including COX4, which was used in our studies. Through this mechanism, the UCP1-deficient adipose tissues exhibit reduced browning phenotype, even though UCP1 is a downstream component. This important issue warrants further in-depth mechanistic studies.

Comment: C2. Kazak and Spiegelman have published numerous studies showing that UCP1 is only one of several futile cycles that leads to thermogenesis in BAT. One notable alternative is futile creatine cycling. Meanwhile, UCP1-KO leads to disrupted mitochondrial structure and function beyond uncoupling and thermogenesis. The authors need additional studies to show that a loss of uncoupling/futile cycles is what leads to the reported anti-tumor effects.

Response: We thank the reviewer for this important comment. Indeed, additional studies to show that a loss of uncoupling/futile cycles is what leads to the reported anti-tumor effects are important. However, genetic deletion of the critical components of the futile creatine cycle is not feasible for us to do at this time. The main reason for not being feasible is that we do not have these genetic models in our laboratory. Importing and obtaining ethical permission for using these genetic models will take very long time (estimated more than one year). We do not exclude the possibility that the futile creatine cycle-related thermogenesis participates in tumor suppression. The central focus of this study is the discovery of a new anticancer drugs.

Genetic deletion of UCP1 already produced marked neutralization of the mirabegron-induced tumor suppression. While this beneficial effect could be related to structural changes of mitochondria, these are positive and rescue experiments. We also backed up with surgical removal of adipose depots. Given the unavailability of these genetic models and focus of this study on cancer suppression, we discussed this issue in the Discussion section. We thank the reviewer for his or her understanding.

Reviewer #2 (Remarks to the Author):

The authors addressed all of my concerns convincingly

Response: We thank the reviewer for satisfying our previous responses.

Reviewer #3 (Remarks to the Author):

Comment: With their revisions, Sun et al have been responsive to the previous round of reviews, significantly increasing the "interpretability" of the manuscript by updating many of the figures to provide actual levels of growth factors. Despite these improvements having removed the human data from this version of the manuscript it is difficult to say if the current version can support the authors largest claims as written.

Response: We thank the reviewer for considering our work significantly improved. To further strengthen our conclusions, we have performed new experiments. We believe that our new data further support our initial conclusions.

Comment: Figure 1 clearly shows that with Mirabegron treatment that several tumor models show decreased growth rates (not as stated inhibition, just decreased growth rates).

Response: We thank the reviewer for this comment. We completely agree. In the revised manuscript, we have used the term "decreased tumor growth rates" to accurately describe our findings.

Comment: Figure 2 demonstrates that Mirabegron treatment increases UCP1 expression in adipose tissues, along with a decrease in adipocyte cell size in WAT and BAT. They show that treatment decreases blood glucose and insulin levels while increasing metabolic rate as gauged by O₂ consumption.

Response: We thank the reviewer for correctly understanding our work.

Comment: Figure 3 they show that if they remove the BAT that they significantly reduce the impact of Mirabegron treatment upon tumor growth, their markers of choice (Ki67, CA9, CC3), and glucose. Despite requests in the last round of review they did not address the changes in insulin nor c-peptide in these models.

Response: We apologize for not including insulin and c-peptide data during the last round of responses. We have performed both non-fasting blood insulin assay and fasting insulin assay in

mirabegron- or vehicle-treated tumor-bearing mice. Non-fasting blood insulin is not altered by mirabegron treatment. Fasting blood insulin in the mirabegron group was reduced by 10% relative the control group (Fig. R8).

Fig. R8.

A. Non-fast and fast blood insulin level of Vehicle- and Mirabegron-treated PDAC (n = 3 mice per group).

We also measured both fasting blood insulin and fasting blood c-peptide. In the vehicle-treated tumor-bearing WT and *Ucp1*^{-/-} mice, no difference in blood insulin levels was observed. However, mirabegron significantly reduced the blood insulin levels in WT mice. Interestingly, knockout of *Ucp1* completely rescued the insulin levels (Fig. R9), suggesting mirabegron-reduced blood insulin is dependent on UCP1.

Blood c-peptide levels in mirabegron- and vehicle-treated WT and *Ucp1*^{-/-} mice were altered in the same manner as blood insulin levels (Fig. R9). Thus, alterations of insulin and c-peptide match each other in our models.

In mirabegron- or vehicle-treated sham-operated or BAT-removed mice, we obtained similar results as *Ucp1*^{-/-} mice (Fig. R9). However, BAT removal did not completely rescue the mirabegron-reduced blood insulin/c-peptide levels (Fig. R9), suggesting that browning WAT may also play role in reducing blood insulin and c-peptide.

These new experimental results have been included in Figs. 2, 3, and 6 of the revised manuscript.

R9

Fig. R9.

- A.** Fast blood insulin and c-peptide level of Vehicle- and Mirabegron-treated CRC tumor-bearing WT and *Ucp1*^{-/-} mice (n = 8 mice per group).
- B.** Fast blood insulin and c-peptide level of sham and BAT removal tumor-bearing mice with or without mirabegron treatment (n = 8 mice per group).

Comment: Figure 4 they argue that Mirabegron treatment inhibits glucose metabolism in the tumor tissue, but they do not control for the impact of insulin/glucose changes in this data set as (again) insulin and glucose levels are associated with the changes they demonstrate in glucose regulating enzymes in the tumor. This feature is not controlled for in their experimental design. This is a key feature since they go on to show that exogenous glucose reduces the impact of Mirabegron in Figure 5. Similarly they show that UCP1 deletion abrogates the impact of Mirabegron in Figure 6.

Response: We thank the reviewer for raising this issue. Indeed, blood glucose and insulin are critical in the glycolytic metabolism in promoting tumor growth. As shown above, since mirabegron downregulates blood insulin levels, we next investigated whether the antitumor effect is insulin-dependent. Vehicle- or mirabegron-treated tumor-bearing mice were treated three times a day with exogenous insulin at the dose of 2 IU/kg. As expected, the administration of insulin has justified to an similar level of the fasting blood insulin in the vehicle- and mirabegron-treated tumor-bearing mice (Fig. R10). Despite the exogenous administration of insulin to reach high blood levels, high blood insulin levels had little impact on tumor growth in the mirabegron-treated tumor-bearing mice (Fig. R10).

R10

Fig. R10.

- A.** Fast blood insulin level at 1h after insulin injection of Vehicle- and Mirabegron-treated PDAC with or without insulin (n = 3 mice per group).
- B.** Tumor growth and tumor inhibition ratios of Vehicle- and Mirabegron-treated PDAC with or without insulin (n = 3 mice per group).

To further elaborate the role of insulin in affecting tumor growth, we performed a loss-of-function experiment by impairing insulin production using streptozotocin (STZ) (45 mg/kg) to treat tumor-bearing mice (Fig. R11). At the end of day 5 after STZ injection, the fasting blood insulin levels were reduced by over 60% (Fig. R11). However, low levels of blood insulin did not retard tumor growth rates, albeit a slightly increased tumor growth rate relative to the control group was observed (Fig. R11). These data indicate that mirabegron-mediated tumor suppression is largely independent from the blood insulin effect.

R11

Fig. R11.

- A.** Diagram of STZ-treated experiment.
- B.** Fast blood insulin level at D-10, D0, and D25 of Vehicle- and STZ-treated CRC-bearing mice (n = 8 mice per group).
- C.** Tumor growth and tumor inhibition ratios of Vehicle- and STZ-treated CRC-bearing mice (n = 9-10 mice per group).

Comment: For both figures 5 and 6 the pre-requisite matched controls are not being run in the same experiment. E.g. in figure 5 we only see mice with 15%HG and no experimental control for the effects is shown within the experiment; similarly there is no syngeneic control for the UCP1 experiments in figure 6. This makes it difficult to gauge the impact of their perturbations as shown/argued.

Response: We apologize for not providing the pre-requisite matched controls for these experiments. In the revised manuscript, proper controls in Fig. 5 and Fig. 6 have been included (Fig. R12).

Fig. R12.

- A.** Tumor growth and tumor inhibition ratios of Vehicle- and Mirabegron-treated CRC tumor-bearing WT and *Ucp1*^{-/-} mice (n = 8-10 mice per group).
- B.** Tumor growth and tumor inhibition ratios of Vehicle- and Mirabegron-treated tumor-bearing mice receiving normal and 15% glucose feeding (n = 8-10 mice per group).

Once again, we thank the reviewers for these valuable and constructive comments which are tremendously helpful for us to improve the quality of our work. We hope that these new experimental data satisfactorily address the reviewers' comments.

Sincerely,

Yihai Cao
 Professor
 Karolinska Institutet
 Email: Yihai.cao@ki.se

REVIEWERS' COMMENTS

Reviewer #1 (Remarks to the Author):

The authors have done an excellent job of responding to the previous set of comments. It is much appreciated that the additional data provided in the rebuttal was then included in the manuscript as these are very important for this manuscript and for moving the field forward.

Reviewer #3 (Remarks to the Author):

In the previous rounds of review, many of my questions/comments were related to the ability of readers/reviewers to assess the data as presented. With the revised manuscript the authors have assuaged many of these concerns. While I do not necessarily agree with their interpretation of the insulin/glucose data, which I think still leaves open the question of the degree to which systemic insulin can be seen as driving some/much of the anti-tumor effects of Mirabegron, as presented I believe that the revised manuscript provides sufficient detail to allow readers to interpret the data for themselves, while making a case for the anti-tumor effects of mirabegron. In all this is a compelling and interesting study that has both therapeutic as well as conceptual innovations.

Manuscript NCOMMS-23-13681-T: Responses to reviewers' comments

Our point-by-point responses to the reviewers' comments are as follows:

Reviewer Comments

Reviewer: 1

Comment: The authors have done an excellent job of responding to the previous set of comments. It is much appreciated that the additional data provided in the rebuttal was then included in the manuscript as these are very important for this manuscript and for moving the field forward.

Response: We thank the reviewer for his or her helpful suggestions, which help us to improve our work.

Reviewer: 3

Comment: In the previous rounds of review, many of my questions/comments were related to the ability of readers/reviewers to assess the data as presented. With the revised manuscript the authors have assuaged many of these concerns. While I do not necessarily agree with their interpretation of the insulin/glucose data, which I think still leaves open the question of the degree to which systemic insulin can be seen as driving some/much of the anti-tumor effects of Mirabegron, as presented I believe that the revised manuscript provides sufficient detail to allow readers to interpret the data for themselves, while making a case for the anti-tumor effects of mirabegron. In all this is a compelling and interesting study that has both therapeutic as well as conceptual innovations.

Response: We thank the reviewer for his or her helpful suggestions, which greatly improve our work. We do agree with the reviewer that our data did not elucidate the degree of systemic insulin in anti-tumor effects of mirabegron. In the revised manuscript, we have further discussed this issue. We thank the reviewer for his or her understanding and for considering our work has therapeutic and conceptual innovations.

Sincerely,

Yihai Cao
Professor
Karolinska Institutet
Email: Yihai.cao@ki.se